# TARFVAE: Efficient One-Step Generative Time Series Forecasting via TARFLOW based VAE

**Jiawen Wei**
Meituan
Beijing, China
weijiawen@meituan.com

**Lan Jiang**
Meituan
Beijing, China
jianglan09@meituan.com

**Pengbo Wei**
Meituan
Beijing, China
weipengbo@meituan.com

**Ziwen Ye**
Meituan
Beijing, China
yeziwen@meituan.com

**Teng Song**
Meituan
Beijing, China
songteng02@meituan.com

**Chen Chen**
Meituan
Beijing, China
chenchen11@meituan.com

**Guangrui Ma**[*]
Meituan
Beijing, China
magr@connect.ust.hk

## Abstract

Time series data is ubiquitous, with forecasting applications spanning from finance to healthcare. Beyond popular deterministic methods, generative models are gaining attention due to advancements in areas like image synthesis and video generation, as well as their inherent ability to provide probabilistic predictions. However, existing generative approaches mostly involve recurrent generative operations or repeated denoising steps, making the prediction laborious, particularly for long-term forecasting. Most of them only conduct experiments for relatively short-term forecasting, with limited comparison to deterministic methods in long-term forecasting, leaving their practical advantages unclear. This paper presents TARFVAE, a novel generative framework that combines the Transformer-based autoregressive flow (TARFLOW) and variational autoencoder (VAE) for efficient one-step generative time series forecasting. Inspired by the rethinking that complex architectures for extracting time series representations might not be necessary, we add a flow module, TARFLOW, to VAE to promote spontaneous learning of latent variables that benefit predictions. TARFLOW enhances VAE's posterior estimation by breaking the Gaussian assumption, thereby enabling a more informative latent space. TARFVAE uses only the forward process of TARFLOW, avoiding autoregressive inverse operations and thus ensuring fast generation. During generation, it samples from the prior latent space and directly generates full-horizon forecasts via the VAE decoder. With simple MLP modules, TARFVAE achieves superior performance over state-of-the-art deterministic and generative models across different forecast horizons on benchmark datasets while maintaining efficient prediction speed, demonstrating its effectiveness as an efficient and powerful solution for generative time series forecasting. Our code is available at https://github.com/Gavine77/TARFVAE.

---

[*]Corresponding author.

39th Conference on Neural Information Processing Systems (NeurIPS 2025).

# 1 Introduction

In various application fields such as transportation planning[1], healthcare[2] and inventory management[3], long-term time series forecasting (LTSF) and uncertainty quantification are of vital importance. The former provides a solid basis for long-term decision-making by offering reliable prediction, while the latter delivers more comprehensive estimates, enabling robust decision-making through accounting for potential variability. Recently, deep learning has achieved remarkable success in these fields through capturing complex time series patterns[4–9] and generating reliable uncertainty estimates[10–13].

In the realm of long-term forecasting, various strategies have been proposed to enhance predictive performance. Most efforts have focused on Transformer-based models due to their proven capability in capturing long-term dependencies via attention mechanisms[6]. Numerous studies aim to reduce the computational complexity of attention[6–8] and improve information extraction, while others apply vanilla Transformers to mine intra-channel or inter-channel relationships[4, 5]. However, some studies[14, 15] argue that sophisticated designs might not be necessary and suggest simple models could deliver results comparable to Transformer-based models. This has spurred exploration into Linear-based or MLP-based methods that aim at more efficient utilization of historical information during training to improve prediction performance[9, 16, 17].

Current LTSF methods are predominantly deterministic, producing point estimates without systematically addressing uncertainty quantification. This oversight poses substantial risks to operational robustness, particularly given the intrinsic positive correlation between uncertainty magnitude and forecast horizon length. To achieve probabilistic forecasting, earlier models like DeepAR[12] employ recurrent neural networks (RNNs) to estimate parameters of prespecified distributions (e.g., Gaussian) for each timestep. However, such methods impose restrictive parametric assumptions that may not capture complex temporal dynamics. Subsequent advancements have explored generative frameworks, including variational autoencoders (VAEs)[18] and diffusion models[10, 13], building upon their demonstrated success in adjacent domains[19–21]. While diffusion models achieve state-of-the-art sample quality, their iterative denoising process incurs substantial computational overhead during inference. Recent attempts to accelerate generation[11, 13, 22] remain fundamentally constrained by the multi-step generation paradigm. By contrast, conventional VAE[23] enables one-step generation by decoding samples from the prior latent space. However, many existing hybrid methods[24–26] introduce complex recurrent structures to capture temporal dependencies and generate predictions autoregressively, which undermine this computational advantage. Due to computational constraints, these generative models have primarily been developed and evaluated in short-term settings, raising concerns about their long-term efficacy and accuracy. Recently, Zhang et al.[27] systematically compared deterministic and probabilistic methods for long-term forecasting, whose results still highlight the need for models that can effectively address both point and probabilistic forecasting across diverse horizons.

Motivated by these considerations, we introduce TARFVAE, a novel VAE-based framework that strategically integrates two core objectives: (1) preserving the one-step generation capability of VAEs to ensure computationally efficient inference, and (2) promoting spontaneous learning of latent variables that guarantee robust probabilistic forecasting performance. TARFVAE achieves these by simply incorporating a Transformer-based autoregressive flow (TARFLOW)[28], which enables an enhanced posterior estimation. Note that the autoregressive inverse process of TARFLOW is not included so TARFVAE can perform one-step generation. Extensive experiments on eight real-world datasets demonstrate the superiority of TARFVAE over the existing state-of-the-art deterministic and generative baselines.

# 2 Related Work

**Long-term Time Series Forecasting**   Transformer-based models have achieved remarkable success in LTSF through their attention mechanisms, which effectively capture long-term dependencies[6–8]. According to Qiu et al.[29], these models can be categorized into channel-independent and channel-dependent approaches. Initially, they aggregate channel information via linear mapping and use attention to extract temporal patterns. However, the high computational complexity of attention mechanisms poses challenges in long-term forecasting. To address this, significant efforts have focused on enhancing attention efficiency[6–8, 30]. Despite progress, channel-dependent

strategies are more vulnerable to distributional shifts[14, 31], prompting some researchers to explore channel-independent approaches, which have shown superior performance[4, 5, 14]. For example, PatchTST[4] splits historical data into patches and applies a standard Transformer to model their correlations. Meanwhile, other studies have leveraged simple Linear-based or MLP-based structures for time series modeling[14, 16, 17, 31]. For instance, SOFTS[17] introduces an MLP-based module to aggregate multiple channels and generate core representations for prediction.

**Generative Models for LTSF**  To boost model robustness and reliability, recent studies have shifted from estimating the median or mean of time series to capturing the complete time series distribution using generative models. Various generative models, such as VAEs[23, 24], normalizing flows[32, 33] and diffusion models[10], have been applied in this field. Transformer-MAF[32] is the pioneering work in applying normalzing flows: it builds upon the Transformer architecture and utilizes Masked Autoregressive Flows (MAF)[34] for generating time series forecasts. Recently, diffusion models have gained prominence in LTSF because of their superior performance in adjacent domains[19–21]. TimeGrad[10], the first time series diffusion model, relied on an RNN to predict in an autoregressive manner, with the denoising process guided by hidden states generated by RNN. Instead of autoregressive decoding, CSDI[11] and SSSD[35] use self-supervised masking to guide the denoising process and make predictions. Methods like TimeDiff[13] and mrDiff[36] utilize different networks to incorporate historical information as conditions and employ conditional diffusion models for time series generation. However, diffusion-based models, though effective, require multiple steps to generate time series from noise, thus slowing inference. Unlike diffusion models, VAEs can efficiently generate data by directly decoding latent representations. Previous studies[18, 24–26] improve the learning of latent space by better representing historical sequences through complex recurrent or autoregressive operations, which compromise VAE's fast inference speed.

## 3 Preliminaries

### 3.1 Variational AutoEncoder

A VAE[23] is an unsupervised generative model that models the input data distribution by learning a probabilistic latent space as follows:

$$p(x) = \int p(x, z)dz = \int p(x|z)p(z)dz = \int p(x|z)\frac{p(z)}{p(z|x)}p(z|x)dz = \mathbb{E}_{z\sim p(z|x)}[p(x|z)\frac{p(z)}{p(z|x)}] \tag{1}$$

where $x$ is the input data and $z$ is its latent representations. The prior $p(z)$ is normally defined as a multivariate Gaussian distribution $N(0, I)$. The posterior $p(z|x)$ can be an arbitrary distribution and VAE approximates it as $q(z|x) = N(\mu(x), \sigma^2(x))$. Thus VAE learns the problem by maximizing the log-likelihood of (1) as

$$\log p(x) = \log \mathbb{E}_{z\sim q(z|x)}\left[p(x|z)\frac{p(z)}{q(z|x)}\right] \tag{2}$$

$$\geq \mathbb{E}_{z\sim q(z|x)}\log\left[p(x|z)\frac{p(z)}{q(z|x)}\right] \tag{3}$$

$$= \mathbb{E}_{z\sim q(z|x)}\left[\log p(x|z) - \log\frac{q(z|x)}{p(z)}\right] \tag{4}$$

$$= \mathbb{E}_{z\sim q(z|x)}\left[\log p(x|z)\right] - KL(q(z|x)||p(z)) \tag{5}$$

where (3) is the evidence lower bound (ELBO) of (2) accoring to Jensen's inequality. The first term in (5) leads to reconstruct accurate $x$ from $z$, and the second term minimizes the difference between the prior and approximated posterior to prompt the latent space to capture meaningful data representations. This enables VAE to generate new samples via $z \sim p(z)$, $x \sim p(x|z)$.

VAE can be extended to conditional VAE by incorporating supervised labels $y$. The ELBO for $\log p(x|y)$ can be analogously derived as follows:

$$\log p(x|y) \geq \mathbb{E}_{z\sim q(z|x,y)}\left[\log p(x|z, y) - \log\frac{q(z|x, y)}{p(z|y)}\right] \tag{6}$$

$$= \mathbb{E}_{z\sim q(z|x,y)}\left[\log p(x|z, y)\right] - KL(q(z|x, y)||p(z|y)) \tag{7}$$

where the prior, approximated posterior and reconstruction process are all conditioned on $y$.

## 3.2 Normalizing Flow

Normalizing flows[33, 37] are invertible mappings $f : \mathcal{X} \to \mathcal{Z}$ from $\mathbb{R}^D$ to $\mathbb{R}^D$. For $x \sim \mathcal{X}$ and $z \sim \mathcal{Z}$, the density $p_{\mathcal{X}}(x)$ can be expressed by

$$p_{\mathcal{X}}(x) = p_{\mathcal{Z}}(f(x)) \left| \det(\frac{\partial f(x)}{\partial x}) \right|. \tag{8}$$

Normalizing flows have the property that the inverse $x = f^{-1}(z)$ is easy to evaluate and computing the Jacobian determinant takes $O(D)$ time. Therefore, once the model is trained, a generative model is automatically obtained via $z \sim \mathcal{Z}$, $x = f^{-1}(z)$. Real NVP[38] designs an affine coupling layer to meet the above two conditions:

$$\begin{cases} z_1 = x_1 \\ z_2 = (x_2 - t(x_1)) \odot \exp(s(x_1)) \end{cases} \tag{9}$$

where $x$ can be randomly shuffled into two parts $[x_1, x_2]$, and $s(x_1)$ and $t(x_1)$ are learnable functions whose outputs match the shape of $x_2$. The inverse of (9) is obvious and the Jacobian matrix of (9) is a lower triangular matrix. The determinant of the Jacobian is the product of the diagonal elements, so the log-determinant is

$$\log |\det \frac{\partial z}{\partial x}| = \sum_i s_i(x_1). \tag{10}$$

To achieve stronger nonlinearity, multiple coupling layers are composed together, creating a chain of mappings: $\mathcal{X} = \mathcal{Z}_0 \to \mathcal{Z}_1 \to \mathcal{Z}_2 \to ... \to \mathcal{Z}_K = \mathcal{Z}$. The maximum likelihood estimation objective can then be written as

$$\log p_{\mathcal{X}}(x) = \log p_{\mathcal{Z}}(z) + \sum_{k=1}^{K} \log |\det \frac{\partial z_k}{\partial z_{k-1}}|. \tag{11}$$

Recently, Zhai et al. proposed TARFLOW[28] which could achieve more efficient nonlinear mappings. TARFLOW partitions $x$ into more parts as $[x_1, x_2, ..., x_n]$ and performs transformation following a similar rule:

$$\begin{cases} z_1 = x_1 \\ z_j = (x_j - t^j(x_{<j})) \odot \exp(s^j(x_{<j})) \end{cases} \tag{12}$$

where $j > 1$ and $x_{<j} = [x_1, x_2, ..., x_{j-1}]$. The $s(x_{<j})$ and $t(x_{<j})$ are efficiently implemented by causal Transformer. Notably, the inverse of (12) becomes

$$\begin{cases} x_1 = z_1 \\ x_j = z_j \odot \exp(-s^j(x_{<j})) + t^j(x_{<j}) \end{cases} \tag{13}$$

where the inference of $x_j$ relies on $x_{<j}$, making the inverse process autoregressive.

## 3.3 Time Series Forecasting

Time series forecasting uses historical time series $x \in \mathbb{R}^{C \times L}$ to predict the future values $y \in \mathbb{R}^{C \times H}$. Here, $C$ represents the number of variables or channels, $L$ is the length of the lookback window, and $H$ stands for the forecast horizon. $C > 1$ corresponds to a multivariate time series; otherwise, it is termed univariate.

Deterministic methods perform point estimation via a function $f$ as $\hat{y} = f(x)$, whereas probabilistic methods model a distribution $\hat{p}(y|x)$ to approximate the true $p(y|x)$.

# 4 TARFVAE

## 4.1 Overview

The architecture of TARFVAE, as illustrated in **Figure 1**, builds upon a conditional VAE where input history $x$ conditions the generation of target series $y$. The Gaussian posterior assumption in Section

3.1 fundamentally limits reconstruction accuracy in conventional VAEs, as Gaussian distributions constitute a tiny subset of all possible posterior distributions. To overcome this expressiveness limitation, we introduce TARFLOW to refine an initial Gaussian posterior $q(z_0|x,y)$ into a flexible complex distribution that closely matches the true posterior. Specifically, both $x$ and $y$ are utilized for training, whereas only $x$ is used for inference.

**Instance normalization**. Following many state-of-the-art models[4, 5, 17, 39], we first apply instance normalization[40] to remove the local statistics of the input history to stabilize the base prediction, and then restore them to the model prediction. Notably, during training, both $x$ and $y$ are normalized based on $x$'s mean and variance without using $y$'s statistics. This ensures $x$'s normalization remains consistent between training and inference, regardless of $y$'s presence.

**Series embedding**. Then we perform series embedding[5, 17] on the needed time series for the encoder, decoder, prior and flow modules independently. Each module employs a dedicated embedding layer to extract module-specific temporal patterns. This architectural isolation prevents cross-module interference while enabling targeted feature learning.

**One-step generation**. The prior and encoder modules parameterize two Gaussians, the prior $p(z|x)$ and initial posterior $q(z_0|x,y)$, respectively. During training, the flow module transforms $z_0 \sim q(z_0|x,y)$ to approximate $z$ from the true posterior, while inference directly samples $z \sim p(z|x)$. Finally, the decoder maps $z$ to $\hat{y}$ conditioned on $x$, enabling efficient one-step full-horizon generation.

The training and inference process of TARFVAE can be formulated as follows:

**Training**

$$\mu_{prior}, \log \sigma^2_{prior} = Prior(Emb_1(x)) \tag{14}$$

$$\mu_{z_0}, \log \sigma^2_{z_0} = Encoder(Emb_2([x,y])) \tag{15}$$

$$z_0 \sim N(\mu_{z_0}, \sigma^2_{z_0}) \tag{16}$$

$$z = TARFLOW(z_0, Emb_3([x,y])) \tag{17}$$

$$\hat{y} = Decoder(z, Emb_4(x)) \tag{18}$$

**Inference**

$$\mu_{prior}, \log \sigma^2_{prior} = Prior(Emb_1(x)) \tag{19}$$

$$z_{prior} \sim N(\mu_{prior}, \sigma^2_{prior}) \tag{20}$$

$$\hat{y} = Decoder(z_{prior}, Emb_4(x)) \tag{21}$$

Below, we describe the TARFVAE implementation used in our work, while noting that alternative approaches could be adopted.

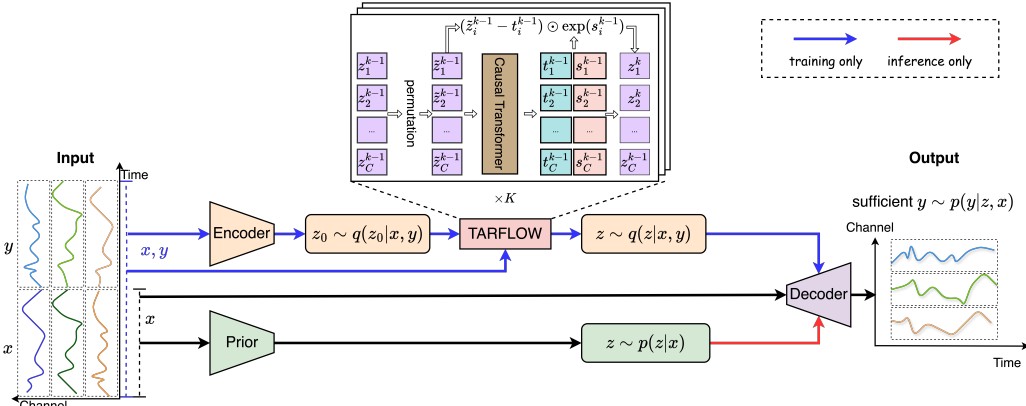

Figure 1: The overview of TARFVAE.

## 4.2 MLP Foundation

Although many Transformer-based architectures such as PatchTST[4], iTranformer[5] and DUET[39] have demonstrated that self-attention can be effective for long-term forecasting, recent studies[14, 15] argue that sophisticated designs might not be necessary and suggest simple models like feedforward neural networks could suffice for the job. To thoroughly test the TARFVAE framework's effectiveness and avoid confounding effects from elaborately designed temporal processing modules, we implement simple MLP blocks as shown in **Figure 2** for basic input mixing, where the channel-wise mappings enable the capture of inter-channel dependencies. These MLP blocks form the architectural foundation across our encoder, decoder, and prior module.

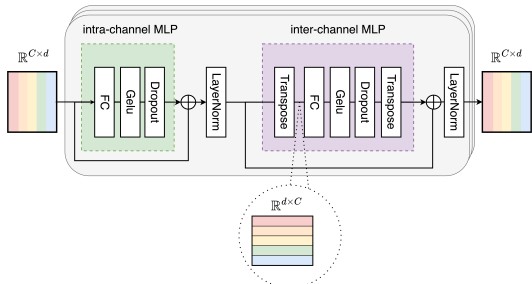

Figure 2: The MLP blocks.

In the prior module and encoder, following multiple MLP blocks, linear layers output the estimated mean and variance for the Gaussian prior $p(z|x)$ and initial posterior $q(z_0|x,y)$ respectively as follows:

$$\mu_{prior}, \log \sigma^2_{prior} = Linear_{prior}(MLPBlocks_{prior}(Emb_1(x))) \tag{22}$$

$$\mu_{z_0}, \log \sigma^2_{z_0} = Linear_{enc}(MLPBlocks_{enc}(Emb_2([x,y]))) \tag{23}$$

The decoder aims to reconstruct $y$ using $z$ conditioned on $x$. We chose to augment $z$ via connecting a full attention output where $z$ queries out relevant information from $x$. Subsequently, MLP blocks process this mixed information, and a linear projection is used to achieve the final reconstruction of y. The process is outlined as follows:

$$h_{mixed} = z + FullAttention(z, Emb_4(x), Emb_4(x)) \tag{24}$$

$$\hat{y} = Linear_{dec}(MLPBlocks_{dec}(h_{mixed})) \tag{25}$$

## 4.3 TARFLOW

In the flow module, the input $z_0$ and all intermediate latent variables $\{z_k\}(k=1,2,...,K)$ maintain consistent dimensionality $\mathbb{R}^{C \times D}$, where $D$ represents the latent dimension. These variables are naturally partitioned by channels, and we perform the transformation (12) conditioned on $x, y$ in each TARFLOW block as follows:

$$\begin{cases} z_1 = x_1 \\ z_j = (x_j - t^j((x + Emb_3([x,y]))_{<j})) \odot \exp(s^j((x + Emb_3([x,y]))_{<j})) \end{cases} \tag{26}$$

Following the original implementation[28], we adopt a dimension-reversing permutation between adjacent blocks. This module finally outputs $z_K \approx z$.

Combining (10), (11) and (6), the training loss for a single sample generation is derived as

$$L = -\mathbb{E}_{z \sim q(z|x,y)} \left[ \log p(y|z,x) - \log q(z_0|x,y) + \sum_{k=1}^{K} \log|\det \frac{\partial z_k}{\partial z_{k-1}}| + \log p(z|x) \right] \tag{27}$$

$$= \frac{D}{2}\ln(2\pi) + \frac{1}{2}\sum_{d=1}^{D}(\ln\sigma^2_{y,d} - \ln\sigma^2_{z_0,d} + \ln\sigma^2_{prior,d}) - \sum_{k=1}^{K}\sum_{d=1}^{D}s_k^d$$

$$+ \frac{1}{2}||\frac{y-\mu_y}{\sigma_y}||^2 - \frac{1}{2}||\frac{z_0 - \mu_{z_0}}{\sigma_{z_0}}||^2 + \frac{1}{2}||\frac{z-\mu_{prior}}{\sigma_{prior}}||^2 \tag{28}$$

where $p(y|z, x)$ is chosen to be Guassian for continuous predictions and its mean $\mu_y$ and variance $\sigma_y^2$ are implemented as $\hat{y}$ and $I$ in our work.

## 5 Experiments

### 5.1 Setup

**Datasets**. To comprehensively evaluate the performance of our proposed TARFVAE, we conduct extensive experiments on 8 widely-used real-world datasets: four ETT subsets (ETTh1, ETTh2, ETTm1, ETTm2), Electricity, Exchange, Weather[6, 7], and Solar-Energy[41].

**Baselines**. We extensively choose the recent state-of-the-art models to serve as baselines. For deterministic methods, we include Linear-based or MLP-based methods (SOFTS[17], TiDE[16], TSMixer[9] and DLinear[14]) and Transformer-based models (DUET[39], iTransformer[5], PatchTST[4], Crossformer[30] and FEDformer[8]). For generative approaches, given their distinct evaluation protocols compared to deterministic approaches, we follow mr-Diff[36] and compare against both mr-Diff and its benchmarked generative baselines: TimeDiff[13], TimeGrad[10], CSDI[11], SSSD[35], D$^3$VAE[22], CPF[42] and PSA-GAN[43].

**Implementation details**. When comparing with deterministic models, the long-term forecasting benchmarks follow the common setting[5–7, 17], with the lookback window length $L$ set to 96 and the prediction horizon $H$ to $\{96, 192, 336, 720\}$ for all datasets. For comparison with mr-Diff and its benchmarked baselines, we adopt the same configurations: $H$ is 168 for Electricity and ETTh1, 192 for ETTm1, and 672 for Weather, while $L$ is chosen from $\{96, 192, 336, 720, 1440\}$. Mean Squared Error (MSE) and Mean Absolute Error (MAE) are adopted as evaluation metrics. Since we can sample different sizes of results once our model is trained, we calculate MSE and MAE for the median of sampled results. We also compute the Continuous Ranked Probability Score (CRPS)[44] based on sampled results as a probabilistic forecasting metric. All our experiments are implemented using PyTorch[45] on a single Nvidia-H20 GPU with 141 GB memory, except for the inference efficiency comparison experiment which is conducted on an Nvidia-A6000 GPU with 48 GB memory to align with the experimental settings of the compared generative baselines. Our training process is guided by the loss function (28) and employs the ADAM optimizer, and the best model is selected based on the MSE of the median of 50 generated samples on the validation set.

### 5.2 Main results

**Table 1** presents the multivariate time series forecasting performance of TARFVAE in contrast with selected deterministic baselines, where the results of sampling 200 are shown and the full results of different sample sizes can be seen in **Table 2**. The results demonstrate that TARFVAE achieves the highest number of best outcomes across various forecast horizons in all 8 datasets. Notably, TARFVAE consistently achieves superior performance under varing forecast horizons in 5 out of 8 datasets with significant improvements. For instance, it reduces average MSE by $4.3\%$ on ETTh2 and $4.8\%$ on Weather, respectively. These results not only demonstrate TARFVAE's precision advantages compared to state-of-the-art deterministic methods but also reveal its ability to produce robust performance in considerable long-term forecasting (e.g., 720-horizon predictions).

**Table 3** shows the performance of TARFVAE compared to other generative methods. TARFVAE achieves the best overall performance, securing top-1 positions in 5 out of 10 metrics and ranking second in 2 metrics. To compare inference efficiency, we rerun TARFVAE under the same configurations as the baselines, since some baselines such as mr-Diff and TimeDiff are not open-sourced. The results shown in **Table 4** demonstrate TARFVAE's significant speed advantages: for instance, it reduces inference time by $85.6\%$ compared to mr-Diff at horizon $H = 720$ in the ETTh1 dataset, and even more so when compared to other methods. Furthermore, TARFVAE maintains computational stability when handling extended forecasting horizons – only the final linear layer's dimensionality increases, while other generative methods require progressive time increments. Additionally, its parallelized sampling computation ensures minimal latency variation (from 1 to 200 samples) under moderate sampling scales. These findings collectively establish TARFVAE as both an accurate and efficient approach, enabling high-quality probabilistic forecasts through one-step generation while outperforming state-of-the-art generative methods in computational scalability.

Table 1: Comparison of multivariate time series forecasting results with selected deterministic baselines across various datasets. We reproduce the results for the Exchange dataset and other results of all baselines are taken from DUET[39] and SOFTS[17].

| Models | | TARFVAE (ours) | | DUET (2025) | | SOFTS (2024) | | iTransformer (2024) | | TSMixer (2023) | | PatchTST (2023) | | Crossformer (2023) | | TiDE (2023) | | DLinear (2023) | | FEDformer (2022) | |
|---|---|---|---|---|---|---|---|---|---|---|---|---|---|---|---|---|---|---|---|---|---|
| Metric | | MSE | MAE | MSE | MAE | MSE | MAE | MSE | MAE | MSE | MAE | MSE | MAE | MSE | MAE | MSE | MAE | MSE | MAE | MSE | MAE |
| ETTh1 | 96 | 0.361 | 0.388 | 0.377 | 0.393 | 0.381 | 0.399 | 0.386 | 0.405 | 0.401 | 0.412 | 0.394 | 0.406 | 0.423 | 0.448 | 0.479 | 0.464 | 0.386 | 0.400 | 0.376 | 0.419 |
| | 192 | 0.410 | 0.422 | 0.429 | 0.425 | 0.435 | 0.431 | 0.441 | 0.436 | 0.452 | 0.442 | 0.440 | 0.435 | 0.471 | 0.474 | 0.525 | 0.492 | 0.437 | 0.432 | 0.420 | 0.448 |
| | 336 | 0.455 | 0.445 | 0.471 | 0.446 | 0.480 | 0.452 | 0.487 | 0.458 | 0.492 | 0.463 | 0.491 | 0.462 | 0.570 | 0.546 | 0.565 | 0.515 | 0.481 | 0.459 | 0.459 | 0.465 |
| | 720 | 0.481 | 0.469 | 0.496 | 0.480 | 0.499 | 0.488 | 0.503 | 0.491 | 0.507 | 0.490 | 0.487 | 0.479 | 0.653 | 0.621 | 0.594 | 0.558 | 0.519 | 0.516 | 0.506 | 0.507 |
| | Avg | 0.427 | 0.431 | 0.443 | 0.436 | 0.449 | 0.443 | 0.454 | 0.448 | 0.463 | 0.452 | 0.453 | 0.446 | 0.529 | 0.522 | 0.541 | 0.507 | 0.456 | 0.452 | 0.440 | 0.459 |
| ETTh2 | 96 | 0.273 | 0.329 | 0.296 | 0.345 | 0.297 | 0.347 | 0.297 | 0.349 | 0.319 | 0.361 | 0.288 | 0.340 | 0.745 | 0.584 | 0.400 | 0.440 | 0.333 | 0.387 | 0.358 | 0.397 |
| | 192 | 0.359 | 0.382 | 0.368 | 0.389 | 0.373 | 0.394 | 0.380 | 0.400 | 0.402 | 0.410 | 0.376 | 0.395 | 0.877 | 0.656 | 0.528 | 0.509 | 0.477 | 0.476 | 0.429 | 0.439 |
| | 336 | 0.391 | 0.409 | 0.411 | 0.422 | 0.410 | 0.426 | 0.428 | 0.432 | 0.444 | 0.446 | 0.440 | 0.451 | 1.043 | 0.731 | 0.643 | 0.571 | 0.594 | 0.541 | 0.496 | 0.487 |
| | 720 | 0.399 | 0.428 | 0.412 | 0.434 | 0.411 | 0.433 | 0.427 | 0.445 | 0.441 | 0.450 | 0.436 | 0.453 | 1.104 | 0.763 | 0.874 | 0.679 | 0.831 | 0.657 | 0.463 | 0.474 |
| | Avg | 0.355 | 0.387 | 0.372 | 0.398 | 0.373 | 0.400 | 0.383 | 0.407 | 0.402 | 0.417 | 0.385 | 0.410 | 0.942 | 0.684 | 0.611 | 0.550 | 0.559 | 0.515 | 0.437 | 0.449 |
| ETTm1 | 96 | 0.311 | 0.351 | 0.324 | 0.354 | 0.325 | 0.361 | 0.334 | 0.368 | 0.323 | 0.365 | 0.329 | 0.365 | 0.404 | 0.426 | 0.364 | 0.387 | 0.345 | 0.372 | 0.379 | 0.419 |
| | 192 | 0.361 | 0.378 | 0.369 | 0.379 | 0.375 | 0.389 | 0.377 | 0.391 | 0.376 | 0.392 | 0.380 | 0.394 | 0.450 | 0.451 | 0.398 | 0.404 | 0.380 | 0.389 | 0.426 | 0.441 |
| | 336 | 0.391 | 0.401 | 0.404 | 0.402 | 0.405 | 0.412 | 0.426 | 0.420 | 0.407 | 0.413 | 0.400 | 0.410 | 0.532 | 0.515 | 0.428 | 0.425 | 0.413 | 0.413 | 0.445 | 0.459 |
| | 720 | 0.456 | 0.435 | 0.463 | 0.437 | 0.466 | 0.447 | 0.491 | 0.459 | 0.485 | 0.459 | 0.475 | 0.453 | 0.666 | 0.589 | 0.487 | 0.461 | 0.474 | 0.453 | 0.543 | 0.490 |
| | Avg | 0.380 | 0.391 | 0.390 | 0.393 | 0.393 | 0.402 | 0.407 | 0.410 | 0.398 | 0.407 | 0.396 | 0.406 | 0.513 | 0.495 | 0.419 | 0.419 | 0.403 | 0.407 | 0.448 | 0.452 |
| ETTm2 | 96 | 0.171 | 0.249 | 0.174 | 0.255 | 0.180 | 0.261 | 0.180 | 0.264 | 0.182 | 0.266 | 0.184 | 0.264 | 0.287 | 0.366 | 0.207 | 0.305 | 0.193 | 0.292 | 0.203 | 0.287 |
| | 192 | 0.229 | 0.291 | 0.243 | 0.302 | 0.246 | 0.306 | 0.250 | 0.309 | 0.249 | 0.309 | 0.246 | 0.306 | 0.414 | 0.492 | 0.290 | 0.364 | 0.284 | 0.362 | 0.269 | 0.328 |
| | 336 | 0.293 | 0.334 | 0.304 | 0.341 | 0.319 | 0.352 | 0.311 | 0.348 | 0.309 | 0.347 | 0.308 | 0.346 | 0.597 | 0.542 | 0.377 | 0.422 | 0.369 | 0.427 | 0.325 | 0.366 |
| | 720 | 0.391 | 0.391 | 0.399 | 0.397 | 0.405 | 0.401 | 0.412 | 0.407 | 0.416 | 0.408 | 0.409 | 0.402 | 1.730 | 1.042 | 0.558 | 0.524 | 0.554 | 0.522 | 0.421 | 0.415 |
| | Avg | 0.271 | 0.316 | 0.280 | 0.324 | 0.288 | 0.330 | 0.288 | 0.332 | 0.289 | 0.333 | 0.287 | 0.330 | 0.757 | 0.611 | 0.358 | 0.404 | 0.350 | 0.401 | 0.305 | 0.349 |
| Exchange | 96 | 0.086 | 0.204 | 0.086 | 0.205 | 0.090 | 0.211 | 0.086 | 0.206 | 0.166 | 0.316 | 0.088 | 0.205 | 0.256 | 0.367 | 0.094 | 0.218 | 0.088 | 0.218 | 0.148 | 0.278 |
| | 192 | 0.172 | 0.294 | 0.182 | 0.305 | 0.182 | 0.304 | 0.177 | 0.299 | 0.279 | 0.402 | 0.176 | 0.299 | 0.470 | 0.509 | 0.184 | 0.307 | 0.176 | 0.315 | 0.271 | 0.315 |
| | 336 | 0.340 | 0.423 | 0.310 | 0.403 | 0.363 | 0.438 | 0.331 | 0.417 | 0.477 | 0.548 | 0.301 | 0.397 | 1.268 | 0.883 | 0.349 | 0.431 | 0.313 | 0.427 | 0.460 | 0.427 |
| | 720 | 0.797 | 0.678 | 0.693 | 0.624 | 0.997 | 0.743 | 0.847 | 0.691 | 0.654 | 0.662 | 0.901 | 0.714 | 1.767 | 1.068 | 0.852 | 0.698 | 0.839 | 0.695 | 1.195 | 0.695 |
| | Avg | 0.349 | 0.400 | 0.318 | 0.384 | 0.408 | 0.424 | 0.360 | 0.403 | 0.394 | 0.482 | 0.367 | 0.404 | 0.940 | 0.707 | 0.370 | 0.414 | 0.354 | 0.414 | 0.519 | 0.429 |
| Electricity | 96 | 0.139 | 0.238 | 0.145 | 0.233 | 0.143 | 0.233 | 0.148 | 0.240 | 0.157 | 0.260 | 0.164 | 0.251 | 0.219 | 0.314 | 0.237 | 0.329 | 0.197 | 0.282 | 0.193 | 0.308 |
| | 192 | 0.159 | 0.255 | 0.163 | 0.248 | 0.158 | 0.248 | 0.162 | 0.253 | 0.173 | 0.274 | 0.173 | 0.262 | 0.231 | 0.322 | 0.236 | 0.330 | 0.196 | 0.285 | 0.201 | 0.315 |
| | 336 | 0.173 | 0.270 | 0.175 | 0.262 | 0.178 | 0.269 | 0.178 | 0.269 | 0.192 | 0.295 | 0.190 | 0.279 | 0.246 | 0.337 | 0.249 | 0.344 | 0.209 | 0.301 | 0.214 | 0.329 |
| | 720 | 0.202 | 0.294 | 0.204 | 0.291 | 0.218 | 0.305 | 0.225 | 0.317 | 0.223 | 0.318 | 0.230 | 0.313 | 0.280 | 0.363 | 0.284 | 0.373 | 0.245 | 0.333 | 0.246 | 0.355 |
| | Avg | 0.168 | 0.264 | 0.172 | 0.259 | 0.174 | 0.264 | 0.178 | 0.270 | 0.186 | 0.287 | 0.189 | 0.276 | 0.244 | 0.334 | 0.252 | 0.344 | 0.212 | 0.300 | 0.214 | 0.327 |
| Solar | 96 | 0.195 | 0.220 | 0.200 | 0.207 | 0.200 | 0.230 | 0.203 | 0.237 | 0.221 | 0.275 | 0.205 | 0.246 | 0.310 | 0.331 | 0.312 | 0.399 | 0.290 | 0.378 | 0.242 | 0.342 |
| | 192 | 0.225 | 0.242 | 0.228 | 0.233 | 0.229 | 0.253 | 0.233 | 0.261 | 0.268 | 0.306 | 0.237 | 0.267 | 0.734 | 0.525 | 0.339 | 0.416 | 0.320 | 0.398 | 0.285 | 0.380 |
| | 336 | 0.250 | 0.262 | 0.262 | 0.244 | 0.243 | 0.269 | 0.248 | 0.273 | 0.272 | 0.294 | 0.250 | 0.276 | 0.750 | 0.735 | 0.368 | 0.430 | 0.353 | 0.415 | 0.282 | 0.376 |
| | 720 | 0.254 | 0.269 | 0.258 | 0.249 | 0.245 | 0.272 | 0.249 | 0.275 | 0.281 | 0.313 | 0.252 | 0.275 | 0.769 | 0.765 | 0.370 | 0.425 | 0.356 | 0.413 | 0.357 | 0.427 |
| | Avg | 0.231 | 0.248 | 0.237 | 0.233 | 0.229 | 0.256 | 0.233 | 0.262 | 0.261 | 0.297 | 0.236 | 0.266 | 0.641 | 0.639 | 0.347 | 0.418 | 0.330 | 0.401 | 0.292 | 0.381 |
| Weather | 96 | 0.151 | 0.197 | 0.163 | 0.202 | 0.166 | 0.208 | 0.174 | 0.214 | 0.166 | 0.210 | 0.176 | 0.217 | 0.158 | 0.230 | 0.202 | 0.261 | 0.196 | 0.255 | 0.217 | 0.296 |
| | 192 | 0.202 | 0.242 | 0.218 | 0.252 | 0.217 | 0.253 | 0.221 | 0.254 | 0.215 | 0.256 | 0.221 | 0.256 | 0.206 | 0.277 | 0.242 | 0.298 | 0.237 | 0.296 | 0.276 | 0.336 |
| | 336 | 0.262 | 0.289 | 0.274 | 0.294 | 0.282 | 0.300 | 0.278 | 0.296 | 0.287 | 0.300 | 0.275 | 0.296 | 0.272 | 0.335 | 0.287 | 0.335 | 0.283 | 0.335 | 0.339 | 0.380 |
| | 720 | 0.342 | 0.341 | 0.349 | 0.343 | 0.356 | 0.351 | 0.358 | 0.347 | 0.355 | 0.348 | 0.352 | 0.346 | 0.398 | 0.418 | 0.351 | 0.386 | 0.345 | 0.381 | 0.403 | 0.428 |
| | Avg | 0.239 | 0.267 | 0.251 | 0.273 | 0.255 | 0.278 | 0.258 | 0.278 | 0.256 | 0.279 | 0.256 | 0.279 | 0.259 | 0.315 | 0.271 | 0.320 | 0.265 | 0.317 | 0.309 | 0.360 |
| $1^{st}$ Count | | 33 | 27 | 2 | 12 | 4 | 2 | 1 | 0 | 1 | 0 | 1 | 1 | 0 | 0 | 0 | 0 | 0 | 0 | 0 | 0 |

Table 2: The inference results of TARFVAE with varing sample sizes: $\{20, 50, 100, 200\}$. The CRPS is calculated using the method in Appendix A.

| Datasets | | ETTh1 | | | ETTh2 | | | ETTm1 | | | ETTm2 | | | Exchange | | | Electricity | | | Solar | | | Weather | | |
|---|---|---|---|---|---|---|---|---|---|---|---|---|---|---|---|---|---|---|---|---|---|---|---|---|---|
| Metric | | MSE | MAE | CRPS | MSE | MAE | CRPS | MSE | MAE | CRPS | MSE | MAE | CRPS | MSE | MAE | CRPS | MSE | MAE | CRPS | MSE | MAE | CRPS | MSE | MAE | CRPS |
| 96 | 20 | 0.376 | 0.399 | 0.323 | 0.275 | 0.331 | 0.302 | 0.321 | 0.359 | 0.296 | 0.177 | 0.257 | 0.215 | 0.087 | 0.206 | 0.176 | 0.143 | 0.243 | 0.203 | 0.201 | 0.224 | 0.183 | 0.154 | 0.200 | 0.173 |
| | 50 | 0.366 | 0.391 | 0.315 | 0.274 | 0.330 | 0.298 | 0.314 | 0.354 | 0.289 | 0.173 | 0.252 | 0.210 | 0.086 | 0.204 | 0.173 | 0.141 | 0.240 | 0.197 | 0.198 | 0.221 | 0.179 | 0.152 | 0.198 | 0.170 |
| | 100 | 0.363 | 0.389 | 0.313 | 0.273 | 0.329 | 0.297 | 0.312 | 0.352 | 0.287 | 0.171 | 0.250 | 0.209 | 0.086 | 0.204 | 0.172 | 0.140 | 0.239 | 0.197 | 0.196 | 0.220 | 0.177 | 0.151 | 0.198 | 0.169 |
| | 200 | 0.361 | 0.388 | 0.311 | 0.273 | 0.329 | 0.296 | 0.311 | 0.351 | 0.286 | 0.171 | 0.249 | 0.208 | 0.086 | 0.204 | 0.171 | 0.139 | 0.238 | 0.196 | 0.195 | 0.220 | 0.177 | 0.151 | 0.197 | 0.169 |
| 192 | 20 | 0.423 | 0.430 | 0.355 | 0.361 | 0.384 | 0.355 | 0.373 | 0.387 | 0.321 | 0.233 | 0.296 | 0.254 | 0.176 | 0.298 | 0.254 | 0.163 | 0.260 | 0.219 | 0.230 | 0.245 | 0.207 | 0.207 | 0.246 | 0.210 |
| | 50 | 0.415 | 0.425 | 0.347 | 0.359 | 0.382 | 0.351 | 0.365 | 0.381 | 0.313 | 0.230 | 0.293 | 0.249 | 0.173 | 0.295 | 0.248 | 0.160 | 0.257 | 0.215 | 0.227 | 0.243 | 0.202 | 0.204 | 0.243 | 0.206 |
| | 100 | 0.412 | 0.423 | 0.344 | 0.359 | 0.382 | 0.350 | 0.363 | 0.379 | 0.311 | 0.229 | 0.292 | 0.247 | 0.172 | 0.294 | 0.246 | 0.159 | 0.256 | 0.213 | 0.226 | 0.243 | 0.201 | 0.203 | 0.242 | 0.204 |
| | 200 | 0.410 | 0.422 | 0.342 | 0.359 | 0.382 | 0.349 | 0.361 | 0.378 | 0.309 | 0.229 | 0.291 | 0.247 | 0.172 | 0.294 | 0.245 | 0.159 | 0.255 | 0.212 | 0.225 | 0.242 | 0.200 | 0.202 | 0.242 | 0.203 |
| 336 | 20 | 0.465 | 0.452 | 0.385 | 0.393 | 0.412 | 0.380 | 0.404 | 0.411 | 0.340 | 0.301 | 0.339 | 0.293 | 0.338 | 0.422 | 0.381 | 0.179 | 0.276 | 0.230 | 0.254 | 0.266 | 0.225 | 0.269 | 0.293 | 0.252 |
| | 50 | 0.458 | 0.447 | 0.378 | 0.391 | 0.410 | 0.375 | 0.395 | 0.404 | 0.332 | 0.295 | 0.335 | 0.288 | 0.339 | 0.422 | 0.376 | 0.175 | 0.272 | 0.224 | 0.252 | 0.263 | 0.220 | 0.264 | 0.290 | 0.246 |
| | 100 | 0.456 | 0.446 | 0.375 | 0.391 | 0.410 | 0.373 | 0.392 | 0.402 | 0.329 | 0.294 | 0.334 | 0.286 | 0.340 | 0.423 | 0.374 | 0.174 | 0.271 | 0.223 | 0.251 | 0.262 | 0.219 | 0.263 | 0.289 | 0.244 |
| | 200 | 0.455 | 0.445 | 0.374 | 0.391 | 0.409 | 0.372 | 0.391 | 0.401 | 0.328 | 0.293 | 0.334 | 0.286 | 0.340 | 0.423 | 0.373 | 0.173 | 0.270 | 0.222 | 0.250 | 0.262 | 0.218 | 0.262 | 0.289 | 0.244 |
| 720 | 20 | 0.502 | 0.483 | 0.414 | 0.403 | 0.431 | 0.394 | 0.470 | 0.445 | 0.373 | 0.400 | 0.397 | 0.345 | 0.792 | 0.676 | 0.647 | 0.208 | 0.299 | 0.253 | 0.258 | 0.272 | 0.234 | 0.349 | 0.345 | 0.300 |
| | 50 | 0.488 | 0.473 | 0.406 | 0.400 | 0.429 | 0.389 | 0.461 | 0.439 | 0.365 | 0.394 | 0.393 | 0.339 | 0.796 | 0.678 | 0.640 | 0.204 | 0.295 | 0.247 | 0.256 | 0.270 | 0.230 | 0.344 | 0.342 | 0.295 |
| | 100 | 0.483 | 0.471 | 0.403 | 0.399 | 0.428 | 0.388 | 0.458 | 0.437 | 0.362 | 0.393 | 0.391 | 0.336 | 0.796 | 0.678 | 0.637 | 0.203 | 0.294 | 0.246 | 0.255 | 0.269 | 0.228 | 0.343 | 0.341 | 0.293 |
| | 200 | 0.481 | 0.469 | 0.401 | 0.399 | 0.428 | 0.387 | 0.456 | 0.435 | 0.360 | 0.391 | 0.391 | 0.335 | 0.797 | 0.678 | 0.636 | 0.202 | 0.294 | 0.245 | 0.254 | 0.269 | 0.227 | 0.342 | 0.341 | 0.292 |
| Avg | 20 | 0.442 | 0.441 | 0.369 | 0.358 | 0.390 | 0.358 | 0.392 | 0.400 | 0.332 | 0.278 | 0.322 | 0.277 | 0.348 | 0.400 | 0.365 | 0.173 | 0.269 | 0.226 | 0.236 | 0.252 | 0.212 | 0.245 | 0.271 | 0.234 |
| | 50 | 0.432 | 0.434 | 0.361 | 0.356 | 0.388 | 0.353 | 0.384 | 0.394 | 0.325 | 0.273 | 0.318 | 0.271 | 0.349 | 0.400 | 0.359 | 0.170 | 0.266 | 0.221 | 0.233 | 0.249 | 0.208 | 0.241 | 0.268 | 0.229 |
| | 100 | 0.428 | 0.432 | 0.359 | 0.356 | 0.387 | 0.352 | 0.381 | 0.392 | 0.322 | 0.272 | 0.317 | 0.270 | 0.349 | 0.400 | 0.357 | 0.169 | 0.265 | 0.219 | 0.232 | 0.249 | 0.206 | 0.240 | 0.267 | 0.228 |
| | 200 | 0.427 | 0.431 | 0.357 | 0.355 | 0.387 | 0.351 | 0.380 | 0.391 | 0.321 | 0.271 | 0.316 | 0.269 | 0.349 | 0.400 | 0.356 | 0.168 | 0.264 | 0.219 | 0.231 | 0.248 | 0.206 | 0.239 | 0.267 | 0.227 |

To ensure a systematic comparison of CRPS performance, we further benchmark our model against the baselines in ProbTS[27] under an identical CRPS protocol. The results in **Table 5** show our TARFVAE still outperforms all probabilistic and point competitors in CRPS except for ranking 2nd in one case, demonstrating the effectiveness of our generative framework. To the best of our knowledge, this work represents the first demonstration of a generative model achieving comprehensive long-term forecasting across diverse datasets (with the number of channels ranging from 7 in ETT to 321 in Electricity) while simultaneously providing state-of-the-art deterministic (MSE, MAE) and probabilistic (CRPS) evaluation metrics.

Table 3: Comparison of multivariate time series forecasting results with selected generative baselines across various datasets. The results of all baselines are taken from mr-Diff[36].

| Datasets | ETTh1 | | ETTm1 | | Exchange | | Electricity | | Weather | | |
|---|---|---|---|---|---|---|---|---|---|---|---|
| Metric | MSE | MAE | MSE | MAE | MSE | MAE | MSE | MAE | MSE | MAE | Avg Rank |
| TARFVAE(ours) | **0.405** (1) | **0.415** (1) | **0.321** (1) | **0.363** (1) | **0.016** (1) | 0.084 (5) | 0.150 (2) | 0.249 (3) | 0.309 (2) | 0.334 (4) | 2.1 |
| mr-Diff | 0.411 (4) | 0.422 (3) | 0.340 (3) | 0.373 (3) | **0.016** (1) | 0.082 (3) | 0.155 (4) | 0.252 (4) | **0.296** (1) | 0.324 (2) | 2.8 |
| TimeDiff | 0.407 (2) | 0.430 (4) | 0.336 (2) | 0.372 (2) | 0.018 (7) | 0.091 (8) | 0.193 (6) | 0.305 (6) | 0.311 (3) | **0.312** (1) | 4.1 |
| TimeGrad | 0.993 (16) | 0.719 (16) | 0.874 (16) | 0.605 (16) | 0.079 (15) | 0.193 (14) | 0.736 (15) | 0.630 (15) | 0.392 (11) | 0.381 (11) | 14.5 |
| CSDI | 0.497 (8) | 0.438 (6) | 0.529 (14) | 0.442 (13) | 0.077 (14) | 0.194 (15) | - | - | 0.356 (9) | 0.374 (9) | 11 |
| SSSD | 0.726 (14) | 0.561 (14) | 0.464 (12) | 0.406 (10) | 0.061 (13) | 0.127 (12) | 0.255 (9) | 0.363 (9) | 0.349 (8) | 0.350 (8) | 10.9 |
| D3VAE | 0.504 (10) | 0.502 (11) | 0.362 (9) | 0.391 (9) | 0.200 (16) | 0.301 (16) | 0.286 (11) | 0.372 (11) | 0.375 (10) | 0.380 (10) | 11.3 |
| CPF | 0.730 (15) | 0.597 (15) | 0.482 (13) | 0.472 (14) | **0.016** (1) | 0.082 (3) | 0.793 (16) | 0.643 (16) | 1.390 (17) | 0.781 (17) | 12.7 |
| PSA-GAN | 0.623 (13) | 0.546 (13) | 0.537 (15) | 0.488 (15) | 0.018 (7) | 0.087 (7) | 0.535 (14) | 0.533 (14) | 1.220 (15) | 0.578 (16) | 12.9 |
| N-Hits | 0.498 (9) | 0.480 (9) | 0.353 (7) | 0.388 (7) | 0.017 (6) | 0.085 (6) | 0.152 (3) | 0.245 (2) | 0.323 (5) | 0.335 (5) | 5.9 |
| FiLM | 0.426 (6) | 0.436 (5) | 0.347 (5) | 0.374 (4) | **0.016** (1) | **0.079** (1) | 0.210 (7) | 0.320 (7) | 0.327 (6) | 0.336 (6) | 4.8 |
| Depts | 0.579 (11) | 0.491 (10) | 0.380 (10) | 0.412 (12) | 0.020 (10) | 0.100 (10) | 0.319 (12) | 0.401 (12) | 0.761 (14) | 0.394 (12) | 11.3 |
| NBeats | 0.586 (12) | 0.521 (12) | 0.391 (11) | 0.409 (11) | **0.016** (1) | 0.081 (2) | 0.269 (10) | 0.370 (10) | 1.344 (16) | 0.420 (13) | 9.8 |
| SCINet | 0.465 (7) | 0.463 (8) | 0.359 (8) | 0.389 (8) | 0.036 (12) | 0.137 (13) | 0.171 (5) | 0.280 (5) | 0.329 (7) | 0.344 (7) | 8 |
| NLinear | 0.410 (3) | 0.418 (2) | 0.349 (6) | 0.375 (5) | 0.019 (9) | 0.091 (8) | **0.147** (1) | **0.239** (1) | 0.313 (4) | 0.328 (3) | 4.2 |
| DLinear | 0.415 (5) | 0.442 (7) | 0.345 (4) | 0.378 (6) | 0.022 (11) | 0.102 (11) | 0.215 (8) | 0.336 (8) | 0.488 (12) | 0.444 (14) | 8.6 |
| LSTMa | 1.149 (17) | 0.782 (17) | 1.030 (17) | 0.699 (17) | 0.403 (17) | 0.534 (17) | 0.414 (13) | 0.444 (13) | 0.662 (13) | 0.501 (15) | 15.6 |

Table 4: The inference time (in ms) of various time series generative models with different prediction lengths on the univariate ETTh1 dataset. The results of all baselines are taken from mr-Diff[36].

| | inference time (ms) | | | |
|---|---|---|---|---|
| | H = 96 | H = 192 | H = 336 | H = 720 |
| TARFVAE (# of samples = 1) | **2.9** | **3.0** | **2.8** | **3.0** |
| TARFVAE (# of samples = 50) | 3.2 | 3.1 | 3.1 | 3.2 |
| TARFVAE (# of samples = 200) | 3.2 | 3.2 | 3.1 | 3.2 |
| mr-Diff(S=2) | 8.3 | 9.8 | 11.9 | 21.6 |
| TimeDiff | 16.2 | 17.6 | 26.5 | 34.6 |
| TimeGrad | 870.2 | 1854.5 | 3119.7 | 6724.1 |
| CSDI | 90.4 | 142.8 | 398.9 | 513.1 |
| SSSD | 418.6 | 645.3 | 1054.2 | 2516.9 |

## 5.3 Ablation study

In this part, we conduct experiments using the exact same configuration as the main experiments, modifying only the targeted settings for analysis to ensure the consistency and reliability of the conclusions.

**Influence of lookback windows**. **Figure 3** demonstrates the performance of TARFVAE with varying lookback window lengths. Overall, TARFVAE consistently achieves superior performance across all evaluated lookback windows, with its accuracy improving as the length gets longer. Notably, on the

Table 5: CRPS comparison following ProbTS[27] protocol: TARFVAE vs. best baselines. The results of all baselines are taken from ProbTS[27].

| Dataset | Pred Len | CRPS | |
|---|---|---|---|
| | | TARFVAE | Other Top Performance |
| ETTm1 | 96 | **0.222** | 0.236 (CSDI) |
| | 192 | **0.244** | 0.291 (CSDI) |
| | 336 | **0.261** | 0.322 (CSDI) |
| | 720 | **0.291** | 0.353 (PatchTST) |
| ETTm2 | 96 | **0.110** | 0.115 (CSDI) |
| | 192 | **0.131** | 0.147 (CSDI) |
| | 336 | **0.150** | 0.176 (PatchTST) |
| | 720 | **0.171** | 0.205 (PatchTST) |
| ETTh1 | 96 | **0.254** | 0.321 (iTransformer) |
| | 192 | **0.278** | 0.359 (iTransformer & PatchTST) |
| | 336 | **0.301** | 0.384 (PatchTST) |
| | 720 | **0.318** | 0.397 (PatchTST) |
| ETTh2 | 96 | **0.150** | 0.164 (CSDI) |
| | 192 | **0.173** | 0.201 (PatchTST) |
| | 336 | **0.185** | 0.240 (PatchTST) |
| | 720 | **0.190** | 0.252 (PatchTST) |
| Electricity | 96 | **0.068** | 0.086 (PatchTST) |
| | 192 | **0.077** | 0.092 (PatchTST) |
| | 336 | **0.081** | 0.099 (GRU NVP) |
| | 720 | **0.087** | 0.108 (TimeGrad) |
| Weather | 96 | **0.065** | 0.068 (CSDI) |
| | 192 | 0.070 | **0.068** (CSDI) |
| | 336 | **0.074** | 0.083 (CSDI) |
| | 720 | **0.080** | 0.087 (CSDI) |
| Exchange | 96 | **0.019** | 0.023 (PatchTST) |
| | 192 | **0.028** | 0.034 (PatchTST) |
| | 336 | **0.041** | 0.048 (iTransformer & PatchTST & DLinear) |
| | 720 | **0.068** | 0.072 (PatchTST) |

ETTm2 dataset, TARFVAE exhibits sustained performance enhancement with increasing window lengths, while competing methods show clear saturation effects.

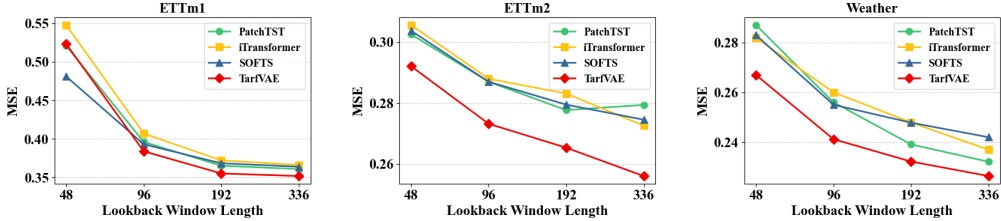

Figure 3: Influence of lookback window length $L$.

Table 6: Ablation study on various datasets. The results are averaged from forecast horizons $H \in \{96, 192, 336, 720\}$ for all datasets with a fixed lookback window length $L = 96$.

| Datasets | ETTh1 | | ETTh2 | | ETTm1 | | ETTm2 | | Exchange | | Electricity | | Solar | | Weather | | AVG | |
|---|---|---|---|---|---|---|---|---|---|---|---|---|---|---|---|---|---|---|
| Metric | MSE | MAE | MSE | MAE | MSE | MAE | MSE | MAE | MSE | MAE | MSE | MAE | MSE | MAE | MSE | MAE | MSE | MAE |
| TARFVAE | **0.432** | **0.434** | **0.356** | **0.388** | **0.384** | **0.394** | **0.273** | **0.318** | **0.347** | **0.307** | **0.170** | **0.266** | **0.233** | **0.297** | **0.241** | **0.268** | **0.301** | **0.333** |
| w/o TARFLOW | 0.441 | 0.441 | 0.379 | 0.404 | 0.396 | 0.402 | 0.279 | 0.324 | 0.420 | 0.323 | 0.180 | 0.278 | 0.316 | 0.300 | 0.243 | 0.270 | 0.332$^{(-10.30\%)}$ | 0.343$^{(-3.00\%)}$ |
| w/o VAE&TARFLOW | 0.466 | 0.457 | 0.386 | 0.410 | 0.394 | 0.407 | 0.306 | 0.342 | 0.409 | 0.322 | 0.186 | 0.283 | 0.275 | 0.298 | 0.249 | 0.276 | 0.334$^{(-10.96\%)}$ | 0.349$^{(-4.80\%)}$ |

**Architecture ablations**. As evidenced in **Table 6**, removing the TARFLOW component and further eliminating VAE lead to a sustained deterioration. These findings collectively demonstrate that the synergistic integration of TARFLOW with VAE critically enables TARFVAE's superior modeling capacity.

**Influence of inference sample size**. Upon completing model training, we evaluate TARFVAE's inference performance under varying sample sizes, as shown in **Table 2**. As the sample size increases from 20 to 200, the reduced sampling bias consistently improves the MSE, MAE and CRPS. The absence of significant metric fluctuations across different sample sizes confirms the stability of TARFVAE's performance. These collective findings substantiate that TARFVAE achieves both reliable uncertainty quantification and deterministic modeling with strong operational stability.

## 6 Conclusion

In this work we present TARFVAE, an efficient one-step generative framework for time series forecasting. It integrates TARFLOW with VAE architecture to enhance posterior estimation, enabling the learning of a more expressive latent space which can lead to efficient modeling of the targeted time series distribution. Implemented on an simple MLP foundation in our extensive experiments, TARFVAE achieves superior or comparable performance to recent state-of-the-art deterministic and generative models across widely-used real-world benchmarks. To our knowledge, this is the first time a generative time series forecasting model delivers comprehensive long-term predictions consistent with deterministic methods on rich datasets, encompassing both deterministic (MSE, MAE) and probabilistic (CRPS) forecasting metrics. Meanwhile, TARFVAE's one-step generation reduces inference time by over $85\%$ compared to other state-of-the-art generative models in long-term forecasting. Other implementations could be tried to achieve better performance in the future.

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

# A Calculation of CRPS

After the inference of sufficient samples, we employ an efficient quantile-based approach to calculate CRPS. This aims to mitigate computation costs in estimating CRPS with all samples. Given certain quantiles $[q_1, q_2, ..., q_n]$, we first sort these samples and extract corresponding quantile values $[Q(q_1), Q(q_2), ..., Q(q_n)]$, then compute CRPS using the following formulation:

$$CRPS = \int_0^1 2 \cdot (Q(q) - y) \cdot (I(y \leq Q(q)) - q) \mathrm{d}q \tag{29}$$

We adopt a constant extrapolation, that is $Q(q) = Q(q_1), \forall q < q_1$ and $Q(q) = Q(q_n), \forall q > q_n$. Thus, (29) can be further formulated as

$$CRPS = \int_0^{q_1} 2 \cdot (Q(q_1) - y) \cdot (I(y \leq Q(q_1)) - q) \mathrm{d}q \tag{30}$$

$$+ \int_{q_1}^{q_n} 2 \cdot (Q(q) - y) \cdot (I(y \leq Q(q)) - q) \mathrm{d}q \tag{31}$$

$$+ \int_{q_n}^1 2 \cdot (Q(q_n) - y) \cdot (I(y \leq Q(q_n)) - q) \mathrm{d}q. \tag{32}$$

As for (30), it equals to

$$\begin{cases} 2 \cdot (Q(q_1) - y) \cdot (q_1 - 0.5q_1^2), & y \leq Q(q_1); \\ 2 \cdot (Q(q_1) - y) \cdot (-0.5q_1^2), & y > Q(q_1). \end{cases} \tag{33}$$

As for (32), it equals to

$$\begin{cases} 2 \cdot (Q(q_n) - y) \cdot (0.5(1 - q_n)^2), & y \leq Q(q_n); \\ 2 \cdot (Q(q_n) - y) \cdot (-0.5(1 - q_n^2)), & y > Q(q_n). \end{cases} \tag{34}$$

As for (31), it is discretized and approximated as

$$2 \cdot \sum_{i=1}^n (Q(q_i) - y) \cdot (I(y \leq Q(q_i)) - q_i) \cdot \Delta q. \tag{35}$$

The inplementation of this calculation process can be found in our code.

# B Limitations and Future Works

We analyze the limitations of our work and briefly discuss several directions for future research. In our work, we employ identical MLP blocks for the encoder, decoder, and prior modules. This configuration might limit the forecasting performance of the proposed TARFVAE model. For instance, since the encoder, decoder, and prior typically perform different functions, we may flexibly design distinct architectures for these three components. Moreover, MLP blocks may not be sufficiently powerful to effectively extract time series patterns from the input data, potentially hindering TARFVAE's ability to learn complex time series patterns. In the future, exploring advanced temporal processing modules or deep learning models within the TARFVAE framework could be beneficial.

# C Societal Impacts

Extensive experiments on real-world datasets demonstrate the TARFVAE's potential capability to significantly benefit various application fields, spanning from finance to energy, by providing accurate and reliable forecasts. However, there are potential negative societal impacts to consider. As a model that often interacts with sensitive and proprietary data, particularly in sectors like healthcare and finance, TARFVAE could inadvertently contribute to privacy risks if the data it processes is not adequately protected. Moreover, Organizations might also become overly reliant on automated forecasts, neglecting valuable insights from human experts, which can affect decision quality. To prevent the aforementioned negative societal impacts, data integrity and security strategies should be well-designed to ensure the deployment of TARFVAE enhances its beneficial societal effects while reducing possible adverse impacts.

