# OpenReview forum: "TARFVAE: Efficient One-Step Generative Time Series Forecasting via TARFLOW based VAE"
_NeurIPS.cc/2025/Conference — NeurIPS 2025 poster_

### Official Review · Reviewer_EWqS · 2025-06-27

**Clarity:** 3
**Significance:** 3
**Originality:** 3
**Rating:** 4
**Confidence:** 3

**Summary:**

This paper proposes TARFVAE, a novel generative framework for time series forecasting that combines Transformer-based autoregressive flow (TARFLOW) with variational autoencoders (VAE). The key innovation is enabling efficient one-step generation for long-term forecasting by using only the forward process of TARFLOW to enhance VAE's posterior estimation, avoiding computationally expensive autoregressive inverse operations while breaking the Gaussian assumption for more informative latent representations.

**Questions:**

1. Why choose MLP decoders over more expressive architectures for complex temporal patterns?
2. How does the method scale with very long input sequences or forecast horizons?
3. The claim that complex architectures aren't necessary needs stronger justification. Have you tested more complex alternatives?

**Ethical Concerns:**

["NO or VERY MINOR ethics concerns only"]

**Final Justification:**

no more questions

**Limitations:**

Yes

**Quality:**

3

**Strengths And Weaknesses:**

Strengths:

1.  The paper addresses a genuine limitation in current generative time series forecasting - the computational inefficiency of recurrent operations and repeated denoising steps.
2. The combination of normalizing flows with VAE for time series forecasting is creative. The specific innovation of using only the forward process of TARFLOW to enhance posterior estimation while avoiding inverse operations is technically sound and addresses the efficiency concerns.
3. The one-step generation capability could be particularly valuable for real-world applications where prediction speed matters.
4. The proposed method outperforms both deterministic and generative SOTA across multiple benchmarks.

Weaknesses:

1. The decision to integrate Transformer architecture specifically within the flow component appears contradictory to the paper's own argument that complex architectures might not be necessary for effective time series representation learning, and the authors should provide more compelling theoretical or empirical justification for this architectural choice.
2. While the paper claims computational efficiency as a major advantage, it would be significantly better to provide comprehensive analysis of the actual time costs for the proposed method compared to baseline methods.
3. The paper fails to adequately explain how the dimensionality of the latent space is determined and whether this dimensionality choice has any principled relationship to the forecast horizon length.

---

> ### Author Rebuttal · Authors · 2025-07-31
>
> Thank you for your thoughtful feedback and recognition of our work. We address your concerns below:
>
> **1.Choice of integrating Transformer into flow while adopting simple MLP encoder/decoder:**
>
> We introduced Transformers specifically in the flow module (TarFlow) rather than the temporal encoder module **to enhance the distribution fitting capabilities**, not to learn complex temporal representations. **We posit that the enhanced posterior distribution approximation enriches the latent space's informational capacity, forming the foundation for improved generation quality, and the powerful distribution transformations performed by TarFlow allows a lightweight encoder.** Therefore, as noted by Reviewer QEJx, we intentionally used a simple MLP backbone to **isolate the contribution of our generative framework and avoid confounding effects from other complex modules**. This design clearly demonstrates the efficacy of our core method. This choice is also inspired by recent works [1,2] suggesting that simple architectures might suffice for learning time series.
>
> **2.Computational efficiency:**
>
> Table 4 empirically compares TARFVAE's inference speed (wall-clock inference time) with generative baselines. Its one-step generation significantly accelerates inference. TARFVAE maintains O(N²) complexity—matching iTransformer and DUET—thus achieving efficiency comparable to these deterministic models.
>
> **3.Latent space dimensionality:**
>
> The latent space dimensionality is treated as a hyperparameter and selected via grid search like other hyperparameters. Theoretically, larger dimensions increase the latent space's informational capacity and may improve generation quality. However, this simultaneously requires the encoder and decoder to enhance their capacity to encode and discriminate latent variables —undoubtedly demanding greater model capacity, more data, and computational resources. Thus, dimensionality selection should balance task complexity, data volume, and computational constraints.
>
> **4.Scaling with lookback/horizon:**
>
> We have reported the model’s performance across datasets as the forecast horizon varies from 96 to 720 in the main results table (**Table 1**), and we illustrate the average performance on each dataset for various forecast horizons while the lookback window is progressively increased from 48 to 336 in **Figure 3**. Overall, as the lookback window lengthens and the forecast horizon shortens, the forecasting task becomes easier and model accuracy improves—consistent with intuition. These results also demonstrate that TARFVAE consistently achieves superior performance compared to baselines across varying lookback windows and forecast horizons.
>
> **5.Alternatives of MLP backbone:**
>
> In response to your interest, **we replace the MLP backbone used in our original experiments with iTransformer** and conduct a quick evaluation on the ETTm and Exchange datasets; the results are reported below. The alternative model delivers performance comparable to the original, still outperforming the other baselines in most cases. Specifically, it enjoys an edge over the original model when the horizon is short (e.g. on ETTm1 and Exchange at horizons of 96 and 192), whereas it falls slightly behind when the horizon is long. This nuanced difference is difficult to interpret conclusively; one plausible explanation is that iTransformer, being more complex than the MLP, induces slight overfitting on longer horizons. At minimum, **the experiment supports the claim that an elaborate backbone is not essential for these tasks: our generative framework powered by Tarflow's potent distribution transformations already suffices to achieve SOTA performance**.
>
> **Table: Results of original TARFVAE and iTransformer alternative. In comparisons between each TARFVAE and the other models, the best result is indicated in bold.**
>
> | Model |  | TARFVAE |  | TARFVAE-iTransformer|  | Other Top Performance |  |
> |:---|:---|:---|:---|:---|:---|:---|:---|
> | Metric |  | MSE | MAE | MSE | MAE | MSE | MAE |
> | ETTm1 | 96 | **0.311** | **0.351** | **0.306** | **0.350** | 0.323(TSMixer) | 0.354(DUET) |
> |  | 192 | **0.361** | **0.378** | **0.356** |**0.378** | 0.369(DUET) | 0.379(DUET) |
> |  | 336 | **0.391** | **0.401** | **0.391** | 0.404 | 0.400(PatchTST) | **0.402(DUET)** |
> |  | 720 | **0.456** | **0.435** | **0.458** | 0.441 | 0.463(DUET) | **0.437(DUET)** |
> |  | Avg | **0.380** | **0.391** | **0.378** | **0.393** | 0.390(DUET) | 0.393(DUET) |
> | ETTm2 | 96 | **0.171** | **0.249** | **0.171** | **0.252** | 0.174(DUET) | 0.255(DUET) |
> |  | 192 | **0.229** | **0.291** | **0.237** | **0.296** | 0.243(DUET) | 0.302(DUET) |
> |  | 336 | **0.293** | **0.334** | **0.301** | **0.341** | 0.304(DUET) | 0.341(DUET) |
> |  | 720 | **0.391** | **0.391** | **0.399** | **0.396** | 0.399(DUET) | 0.397(DUET) |
> |  | Avg | **0.271** | **0.316** | **0.277** | **0.321** | 0.280(DUET) | 0.324(DUET) |
> | Exchange | 96 | **0.086** | **0.204** | **0.083** | **0.202** | 0.086(DUET & iTransformer) | 0.205(DUET & PatchTST) |
> |  | 192 | **0.172** | **0.294** | **0.170** | **0.294** | 0.176(PatchTST & DLinear) | 0.299(iTransformer & PatchTST) |
> |  | 336 | 0.340 | 0.423 | 0.320 | 0.410 | **0.301(PatchTST)** | **0.397(PatchTST)** |
> |  | 720 | 0.797 | 0.678 | 0.855 | 0.691 | **0.654(TSMixer)** | **0.624(DUET)** |
> |  | Avg | 0.349 | 0.400 | 0.357 | 0.399 | **0.318(DUET)** | **0.384(DUET)** |
>
>
> Reference:
>
> [1] Zeng et al. "Are transformers effective for time series forecasting? ". AAAI 2023.
>
> [2] Sun et al. "Simple feedfoward neural networks are almost all you need for time series forecasting". 2025.

---

### Official Review · Reviewer_3TPd · 2025-07-01

**Clarity:** 3
**Significance:** 3
**Originality:** 2
**Rating:** 5
**Confidence:** 3

**Summary:**

The paper proposes to treat time series prediction as conditional generation: given a lookback history, e.g. 96 consecutive observations, one may predict the next observations up to a prespecified horizon e.g. 192 observations. In such a setting, one may use the conditional VAE architecture where the historical observations are given as conditioning input, this defines a distribution in latent space from which we sample and decode to produce the full matrix of horizon predictions in one neural network pass. Sampling multiple latents and decoding yields multiple output predictions.

This is contrast to autoregressive methods that must compute predictions up to a horizon time step by time step each requiring a network pass, or diffusion that requires iterative denoising where each denoising step is a single network pass.

At a high level the method uses the standard C-VAE components, similar to a standard VAE however a conditioning  variable informs the prior over latent variable and the likelihood (decoder) and also the approximate posterior over the latent variable (encoder). Typically the approximate posterior is takes observed data as input and returns an isotrpoic Gaussian distribution, this work uses a Normalizing Flow model that incorporates a transformer, TARFLOW. During training, the past data and future data is passed to the encoder, the TARFLOW approximate posterior, the ELBO is used to inform the approximate posterior to align with the Gaussian prior (determined lookback data) and the likelihood of the output data (decoder reconstruction).

At inference time, the lookback data is used to inform the Gaussian prior over latent variables, a latent vector is sampled, and the lookback data as well as the sampled latent value are decoded to yield a matrix of predictions for the time series up to the horizon. The TARFLOW model is not required.

Experiments are performed on a range of datasets from the literature and show favorable performance against a range of recent methods from the literature.

**Questions:**

- how to handle test time sequences shorter than the lookback duration?
- can the authors report mean MSE and MAE in line with the results copied from SOFTS and DUET?
- can the authors elaborate on why a Conditional-VAE outperforms diffusion somewhat opposing the community consensus that diffusion

**Ethical Concerns:**

["NO or VERY MINOR ethics concerns only"]

**Final Justification:**

All of my issues have been resolved. My main concern was the reported metric for which I had partly misunderstood and has been satisfactoraly cleared up.

**Limitations:**

- input and outpout shapes are fixed during training (which is true for some diffusion methods too)

**Paper Formatting Concerns:**

Not that I noticed.

**Quality:**

3

**Strengths And Weaknesses:**

# Strengths
- the approach seems simple, elegant, a Conditoinal VAE that uses modern architectures.

# Weaknesses
- fixed time length, auto regressive for longer time series than the model was trained for still requires multiple forward passes, can the model be used for settings where the lookback is shorter than whaat was trained? (Presumably the same limitation applies to diffusion based methods)
- where is the baseline mrDIFF?
- it seems the SOFTS and DUET authors report the mean of MSE and MAE of predictions, whereas this work shows the median. This makes the results not comparable (arguably median may be lower as it negates large MSE outliers and MSE>0 bounds low MSE outliers). If the mean MSE, MAE are still outperforming baselines, then I will happily raise my score.

## Minor Comment
- is it possible to see the MSE per time step? Does the MSE grow with time?
- the hypothesis of the paper seems strange. In generative modeling, in the last few years, iterative refinement methods, using autoregression or diffusion, have been outperforming one-shot methods like VAEs and GANs. This paper argues that the Gaussian approximate posterior is to blame and simply making the approximate posterior more flexible during training gives it the advantage. I realise this is high level and speculative and the ablation study removing TARFLOW shows a decrement, but it may be worth acknowledging or discussing and make the paper more convincing: a more flexible approximate posterior can make VAEs outperform diffusion apparently.
- the notation switch of learning $(x,y)$ in Sectoin 3 to $(x, y)$ in section 4 was rather confusing at first

---

> ### Author Rebuttal · Authors · 2025-07-31
>
> Thank you for the helpful comments. Below we address each concern:
>
> **1. Variable-length lookback / horizon:**
>
> Like most LTSF models, ours is trained with fixed lengths. For test sequences shorter than the trained lookback window, common solutions include:
>
> - retraining on the desired window;
>
> - padding (e.g., padding at the beginning, as in Crossformer[1]);
>
> - adopting placeholder tokens and masking mechanisms for variable-length inputs (e.g., MOIRAI [2], ElasTST [3]).
>
> Our framework can readily incorporate similar mechanisms for flexible forecasting.
>
> **2. mr-DIFF baseline:**
>
> Results for mr-DIFF are included in Table 3 (accuracy) and Table 4 (inference speed). We confirm no baseline was omitted.
>
> **3. Metric reporting (Mean vs. Median):**
>
> We clarify that our reported MAE/MSE values are averages over the entire forecast horizon, identical to SOFTS and DUET. The term **'median' refers only to the median over multiple stochastically generated samples** of the same model (i.e., we report the performance of median across generated samples). Table 2 demonstrates the robustness of this sample median. For direct comparison, we are willing to **report the corresponding mean-over-samples scores below**. The results demonstrate that while the mean performance is marginally weaker than the median (sometimes also better, e.g. Exchange 192), they remain closely aligned. Nevertheless, **both significantly outperform other baseline models**.
>
> **Table A: TARFVAE's results of median (the original) and mean over the same generated samples. In comparisons between each TARFVAE and the other models, the best result is indicated in bold.**
> | Model |  | TARFVAE |  | TARFVAE-mean |  | Other Top Performance |  |
> |:---|:---|:---|:---|:---|:---|:---|:---|
> | Metric |  | MSE | MAE | MSE | MAE | MSE | MAE |
> | ETTh1 | 96 | **0.360** | **0.388** | **0.360** | **0.388** | 0.376(FEDformer) | 0.393(DUET) |
> |  | 192 | **0.410** | **0.422** | **0.410** | **0.423** | 0.420(FEDformer) | 0.425(DUET) |
> |  | 336 | **0.455** | **0.445** | **0.455** | **0.446** | 0.459(FEDformer) | 0.446(DUET) |
> |  | 720 | **0.481** | **0.469** | **0.484** | **0.470** | 0.487(PatchTST) | 0.479(PatchTST) |
> |  | Avg | **0.427** | **0.431** | **0.427** | **0.432** | 0.440(FEDformer) | 0.436(DUET) |
> | ETTh2 | 96 | **0.273** | **0.329** | **0.273** | **0.329** | 0.288(PatchTST) | 0.340(PatchTST) |
> |  | 192 | **0.359** | **0.383** | **0.359** | **0.382** | 0.368(DUET) | 0.389(DUET) |
> |  | 336 | **0.391** | **0.409** | **0.392** | **0.410** | 0.410(SOFTS) | 0.422(DUET) |
> |  | 720 | **0.399** | **0.428** | **0.399** | **0.428** | 0.411(SOFTS) | 0.433(SOFTS) |
> |  | Avg | **0.356** | **0.387** | **0.356** | **0.387** | 0.372(DUET) | 0.398(DUET) |
> | ETTm1 | 96 | **0.311** | **0.351** | **0.312** | **0.352** | 0.323(TSMixer) | 0.354(DUET) |
> |  | 192 | **0.361** | **0.378** | **0.362** | **0.378** | 0.369(DUET) | 0.379(DUET) |
> |  | 336 | **0.391** | **0.401** | **0.391** | **0.401** | 0.400(PatchTST) | 0.402(DUET) |
> |  | 720 | **0.456** | **0.435** | **0.457** | **0.436** | 0.463(DUET) | 0.437(DUET) |
> |  | Avg | **0.380** | **0.391** | **0.380** | **0.392** | 0.390(DUET) | 0.393(DUET) |
> | ETTm2 | 96 | **0.171** | **0.249** | **0.171** | **0.250** | 0.174(DUET) | 0.255(DUET) |
> |  | 192 | **0.229** | **0.291** | **0.230** | **0.293** | 0.243(DUET) | 0.302(DUET) |
> |  | 336 | **0.293** | **0.334** | **0.296** | **0.336** | 0.304(DUET) | 0.341(DUET) |
> |  | 720 | **0.391** | **0.391** | **0.397** | **0.394** | 0.399(DUET) | 0.397(DUET) |
> |  | Avg | **0.271** | **0.316** | **0.273** | **0.318** | 0.280(DUET) | 0.324(DUET) |
> | Exchange | 96 | **0.086** | **0.204** | **0.086** | **0.204** | **0.086(DUET & iTransformer)** | 0.205(DUET & PatchTST) |
> |  | 192 | **0.172** | **0.294** | **0.170** | **0.293** | 0.176(PatchTST & DLinear) | 0.299(iTransformer & PatchTST) |
> |  | 336 | 0.340 | 0.423 | 0.327 | 0.416 | **0.301(PatchTST)** | **0.397(PatchTST)** |
> |  | 720 | 0.797 | 0.678 | 0.822 | 0.689 | **0.654(TSMixer)** | **0.624(DUET)** |
> |  | Avg | 0.349 | 0.400 | 0.351 | 0.400 | **0.318(DUET)** | **0.384(DUET)** |
> | Electricity | 96 | **0.139** | 0.238 | **0.140** | 0.239 | 0.143(SOFTS) | **0.233(DUET & SOFTS)** |
> |  | 192 | 0.159 | 0.255 | 0.161 | 0.257 | **0.158(SOFTS)** | **0.248(DUET & SOFTS)** |
> |  | 336 | **0.173** | 0.270 | 0.176 | 0.271 | 0.175(DUET) | **0.262(DUET)** |
> |  | 720 | **0.202** | 0.294 | 0.206 | 0.295 | 0.204(DUET) | **0.291(DUET)** |
> |  | Avg | **0.168** | 0.264 | **0.171** | 0.265 | 0.172(DUET) | **0.259(DUET)** |
> | Solar | 96 | **0.195** | 0.220 | **0.195** | 0.224 | 0.200(DUET & SOFTS) | **0.207(DUET)** |
> |  | 192 | **0.225** | 0.242 | **0.227** | 0.245 | 0.228(DUET) | **0.233(DUET)** |
> |  | 336 | 0.250 | 0.262 | 0.254 | 0.264 | **0.243(SOFTS)** | **0.244(DUET)** |
> |  | 720 | 0.254 | 0.269 | 0.257 | 0.270 | **0.245(SOFTS)** | **0.249(DUET)** |
> |  | Avg | 0.231 | 0.248 | 0.233 | 0.251 | **0.229(SOFTS)** | **0.233(DUET)** |
> | Weather | 96 | **0.151** | **0.197** | **0.155** | **0.201** | 0.158(Crossformer) | 0.202(DUET) |
> |  | 192 | **0.202** | **0.242** | 0.209 | **0.247** | 0.206(Crossformer) | 0.252(DUET) |
> |  | 336 | **0.262** | **0.289** | **0.270** | **0.294** | 0.272(Crossformer) | 0.294(DUET) |
> |  | 720 | **0.342** | **0.341** | **0.347** | **0.343** | 0.349(DUET) | 0.343(DUET) |
> |  | Avg | **0.239** | **0.267** | **0.245** | **0.271** | 0.251(DUET) | 0.273(DUET) |
> | Count 1st |  | 33 | 27 | 30 | 27 |  |  |
>
> **4. Per-timestep MSE:**
>
> As requested, we **provide some detailed per-timestep MSE (see below)**. As expected, MSE generally increases with prediction horizon (periodic fluctuations may occur in datasets with inherent seasonality, such as ETTm1, which can be easily discerned from the plot), aligning with the intuition that distant forecasts are more challenging.
>
> **Table B: Detailed per-timestep MSE. We take Electricity, Exchange and ETTm1 with horizon=720 for example, and report results every 50 steps to shorten the table.**
> | Step (of horizon=720) | Electricity | Exchange | ETTm1 |
> |:---|:---|:---|:---|
> | 1 | 0.088 | 0.023 | 0.075 |
> | 51 | 0.145 | 0.114 | 0.35 |
> | 101 | 0.157 | 0.223 | 0.338 |
> | 151 | 0.171 | 0.355 | 0.404 |
> | 201 | 0.19 | 0.481 | 0.399 |
> | 251 | 0.189 | 0.587 | 0.442 |
> | 301 | 0.196 | 0.705 | 0.47 |
> | 351 | 0.214 | 0.83 | 0.502 |
> | 401 | 0.22 | 0.938 | 0.492 |
> | 451 | 0.217 | 1.041 | 0.503 |
> | 501 | 0.219 | 1.083 | 0.523 |
> | 551 | 0.232 | 1.133 | 0.541 |
> | 601 | 0.242 | 1.22 | 0.543 |
> | 651 | 0.251 | 1.387 | 0.54 |
> | 701 | 0.266 | 1.489 | 0.568 |
>
> **5. Why C-VAE:**
>
> While iterative refinement methods like diffusion gained popularity for high-fidelity generation, their inherently iterative nature incurs **substantial computational costs**. This has **spurred renewed interest in one-step generation** paradigms. Recent studies (e.g., Mean Flows[4], OSV[5]) ultilize pretrained VAEs/GANs and achieve high-quality image/video generation through efficient latent-space discrimination in a single step. Their success suggests that **well-structured latent spaces may inherently contain sufficient information for high-quality generation**. When unlocked through task-specific constraints (e.g., flow matching identities, adversarial discriminators), the information enables high-fidelity one-step mapping.
>
> Similarly, for our proposed TARFVAE, we posit that **Transformer-based autoregressive flows (TarFlow) enhances the VAE encoder by enriching the latent space's informational capacity**, forming the foundation for improved quality. This also **boosts the decoder to specialize in latent-variable discrimination and master one-step mapping** (since VAE encoder and decoder implicitly learn in an inherent adversarial way). Compared to diffusion models where each step approximates a local Gaussian assumption, we posit that an one-step model can match or surpass diffusion performance — **provided the sufficient expressiveness of latent space (always non-Gaussian) and potent latent-variable discrimination, the information flux of a single step can compete that accumulated over multiple steps**.
>
> Reference:
>
> [1] Yunhao Zhang, Junchi Yan. "Crossformer: Transformer Utilizing Cross-Dimension Dependency for Multivariate Time Series Forecasting". ICLR 2023.
>
> [2] Woo et al. "Unified Training of Universal Time Series Forecasting Transformers". PMLR 2024.
>
> [3] Jiawen Zhang. "ElasTST: Towards Robust Varied-Horizon Forecasting with Elastic Time-Series Transformer". NeurIPS 2024.
>
> [4] Zhengyang Geng et al. "Mean Flows for One-step Generative Modeling". 2025.
>
> [5] Mao et al. "OSV: One Step is Enough for High-Quality Image to Video Generation". CVPR 2025.

---

> ### Comment · Area_Chair_2gD4 · 2025-08-05
> **Please respond to authors' rebuttal before Aug. 6 (AOE)**
>
> Dear Reviewer 3TPd,
>
> This is a reminder that the author-reviewer discussion period is ending soon on Aug. 6 (AOE), and you have not yet responded to authors' rebuttal. Please read authors' rebuttal as soon as possible, and engage in any necessary discussions, and consider if you would like to update your review and score. Please at least submit the Mandatory Acknowledgement as a sign that you have completed this task.
>
> Thank you for your service in the review process.
>
> AC

---

> ### Comment · Reviewer_3TPd · 2025-08-05
> **thank you for the thorough response**
>
> All my points have been addressed and I have raised my score.

---

> > ### Author Response · Authors · 2025-08-05
> > **Response to Reviewer 3TPd: Thank You**
> >
> > Dear Reviewer 3TPd,
> >
> > Thank you very much for your positive feedback and for raising your score. We sincerely appreciate the time and valuable insights you dedicated to reviewing our work and are delighted that our responses addressed your points satisfactorily.
> >
> > Best regards,
> >
> > Paper 28044 Authors

---

### Official Review · Reviewer_zanJ · 2025-07-03

**Clarity:** 3
**Significance:** 2
**Originality:** 2
**Rating:** 4
**Confidence:** 3

**Summary:**

The work proposes TARFVAE, an approach that integrates transformer-based auto-regressive flows (TARF) with VAE, for time series forecasting task. It uses TARFs to model variational posterior latent distributions which are more expressive than parameterized Gaussian distributions, which is typically used in VAE. The approach also permits one-step generation for future forecasting, achieving great computational efficiency. As a generative model, the proposed approach shows performance and efficiency superior to both deterministic and generative models on various time series datasets, particularly in long-term forecasting.

**Questions:**

Please see the weakness section.

**Ethical Concerns:**

["NO or VERY MINOR ethics concerns only"]

**Final Justification:**

I'm still not completely satisfied with the limited novelty and plain contributions of the work. Overall speaking, the work is much more solid after my concerns were addressed by the author.

**Limitations:**

yes

**Paper Formatting Concerns:**

I have no major concerns with the formatting of the paper but I would suggest the author use the space more efficiently when displaying the math equations. For example, Equations 2 - 7 and Equation 14 -19 can be displayed in the paragraph's text instead of being displayed in a new line.

**Quality:**

3

**Strengths And Weaknesses:**

## Strengths:
1. Experiment results on standard benchmark datasets outperformed the SOTA models while preserving generation capability and great computational efficiency due to one-step generation. The experiment results provides solid support to the claimed contributions of the work.
2. The work provides rich technical details when describing the TARFVAE method. The presentation is easy to follow and read.

## Weakness:
1. My biggest concern of the work is the limited novelty despite its superior performance and computational efficiency. The approach of using normalizing flows to model the variational posterior distribution in VAE has been explored in previous works [1]. The work can be positioned as an work that extend this approach to the time-series domain.
2. The major experiment results are based on an MLP architecture proposed in the work. Exploring other model architectures like transformers could both highlight the generalizability and flexibility of the work and makes the experiment results more solid.

References:

[1] Rezende, Danilo, and Shakir Mohamed. "Variational inference with normalizing flows." ICML 2015

---

> ### Author Rebuttal · Authors · 2025-07-31
>
> Thank you for the detailed comments. We appreciate your concerns and address them as follows:
>
> **1.Novelty through Cross-Domain Innovation & Advancement:**
>
> While [1] pioneered flows in VAEs, our work **introduces key innovations**:
>
> - further employing Transformer-based autoregressive flows (TarFlow) to strenthen VAE posterior approximation;
>
> - strategic use of TarFlow's unidirectional flow enabling efficient one-step generation without compromising expressiveness. This avoids the costly bijective inversion at inference, yielding a significant efficiency gain over standard flow-VAEs.
>
> - first application of flow-VAE integration to generative time-series forecasting.
>
> Cross-domain adaptation is fundamental to research progress. As Transformers migrated from NLP to CV, AI and beyond, enabling new capabilities and deeper understanding, our work **demonstrates how advancing flow-VAEs in time-series**:
>
> - solves unique challenges (long-horizon forecasting);
>
> - reveals new efficiency-performance tradeoffs (one-step generation with high quality);
>
> - establishes competitive alternatives to domain-specific architectures (SOTA performance).
>
> Our work **aligns with recent related efforts to re-explore underutilized previous ideas**:
>
> - A decade after [1], [2] revisited flows and pretrained VAEs for efficient one-step image generation.
>
> - Despite diffusion models' dominance, TarFlow[3] revived interest in flows by demonstrating state-of-the-art image generation (though building on IAF[4]).
>
> - While diffusion models and recurrent VAEs dominate generative time-series forecasting via computationally intensive multi-step generation, our TARFVAE framework achieves superior efficiency and performance through one-step generation.
>
> **2.MLP Architecture Choice:**
>
> As noted by Reviewer QEJx, we intentionally used a simple MLP backbone to isolate the contribution of our generative framework and avoid confounding effects from other complex modules. This design clearly demonstrates the efficacy of our core method.
>
> Following your suggestion, **we replace the MLP backbone used in our original experiments with iTransformer** and conduct a quick evaluation on the ETTm and Exchange datasets; the outcomes are shown below. The new configuration matches the original in overall quality, continuing to surpass other competitors in most settings. Specifically, it enjoys an edge over the original model when the horizon is short (e.g. on ETTm1 and Exchange at horizons of 96 and 192), whereas it falls slightly behind when the horizon is long. This nuanced difference is difficult to interpret conclusively; one plausible explanation is that iTransformer, being more complex than the MLP, induces slight overfitting on longer horizons. Taken together, these findings further **underscore the generalizability and flexibility of our approach and suggest that the proposed generative framework empowered by TarFlow’s potent distribution transformations already suffices to achieve state-of-the-art performance**.
>
> **Table: Results of original TARFVAE and iTransformer alternative. In comparisons between each TARFVAE and the other models, the best result is indicated in bold.**
>
> | Model |  | TARFVAE |  | TARFVAE-iTransformer|  | Other Top Performance |  |
> |:---|:---|:---|:---|:---|:---|:---|:---|
> | Metric |  | MSE | MAE | MSE | MAE | MSE | MAE |
> | ETTm1 | 96 | **0.311** | **0.351** | **0.306** | **0.350** | 0.323(TSMixer) | 0.354(DUET) |
> |  | 192 | **0.361** | **0.378** | **0.356** |**0.378** | 0.369(DUET) | 0.379(DUET) |
> |  | 336 | **0.391** | **0.401** | **0.391** | 0.404 | 0.400(PatchTST) | **0.402(DUET)** |
> |  | 720 | **0.456** | **0.435** | **0.458** | 0.441 | 0.463(DUET) | **0.437(DUET)** |
> |  | Avg | **0.380** | **0.391** | **0.378** | **0.393** | 0.390(DUET) | 0.393(DUET) |
> | ETTm2 | 96 | **0.171** | **0.249** | **0.171** | **0.252** | 0.174(DUET) | 0.255(DUET) |
> |  | 192 | **0.229** | **0.291** | **0.237** | **0.296** | 0.243(DUET) | 0.302(DUET) |
> |  | 336 | **0.293** | **0.334** | **0.301** | **0.341** | 0.304(DUET) | 0.341(DUET) |
> |  | 720 | **0.391** | **0.391** | **0.399** | **0.396** | 0.399(DUET) | 0.397(DUET) |
> |  | Avg | **0.271** | **0.316** | **0.277** | **0.321** | 0.280(DUET) | 0.324(DUET) |
> | Exchange | 96 | **0.086** | **0.204** | **0.083** | **0.202** | 0.086(DUET & iTransformer) | 0.205(DUET & PatchTST) |
> |  | 192 | **0.172** | **0.294** | **0.170** | **0.294** | 0.176(PatchTST & DLinear) | 0.299(iTransformer & PatchTST) |
> |  | 336 | 0.340 | 0.423 | 0.320 | 0.410 | **0.301(PatchTST)** | **0.397(PatchTST)** |
> |  | 720 | 0.797 | 0.678 | 0.855 | 0.691 | **0.654(TSMixer)** | **0.624(DUET)** |
> |  | Avg | 0.349 | 0.400 | 0.357 | 0.399 | **0.318(DUET)** | **0.384(DUET)** |
>
>
> Reference:
>
> [1] Rezende, Danilo, and Shakir Mohamed. "Variational inference with normalizing flows." ICML 2015.
>
> [2] Zhengyang Geng et al. "Mean Flows for One-step Generative Modeling". 2025.
>
> [3] Zhai et al. "Normalizing Flows are Capable Generative Models". ICML 2025.
>
> [4] Kingma et al. "Improving Variational Inference with Inverse Autoregressive Flow". NIPS 2016.

---

> ### Comment · Reviewer_zanJ · 2025-08-04
> **Updates after rebuttal.**
>
> I would like to thank the author for the additional study on TARFVAE. However, my concerns on the lack of original contributions of novelty was not fully addressed for the following two reasons:
> 1.  The key contribution of the work is the introduction of TaRFlow. One-step generation and inference speed is claimed as one of the major benefits of the work. In Table 4, the author cites the inference time of baseline models from another work. This might not be a controlled experiment considering the experiment settings, including the hardwares that are used to time the inference, could be difference.
> 2. There are major other potential benefits of this approach left unexplored, including variable-length lookback / horizon in the response to Reviewer 3TPd.

---

> > ### Author Response · Authors · 2025-08-04
> > **Clarifying and discussion**
> >
> > Thank you for the careful follow-up.
> >
> > Regarding the three points you raised:
> > - We respectfully draw your attention to **p. 8**, where we explicitly state that *“we rerun TARFVAE **under the same configurations as the baselines**, since some baselines such as mr-Diff and TimeDiff are not open-sourced”*. In addition, **p. 7 (Implementation Details)** clarifies that *“the inference efficiency comparison experiment which is conducted **on an Nvidia-A6000 GPU with 48 GB memory to align with the experimental settings of the compared generative baselines**”*. Therefore, the reported times are obtained **under identical hardware and software settings**, ensuring a controlled and fair comparison.
> > - While extending to variable-length settings is valuable, the majority of recent work in this line (including TARFVAE, mr-Diff, TimeDiff) evaluates on fixed lookback/horizon to isolate the contribution of the generative mechanism itself.
> > In our response to Reviewer 3TPd we already sketched how TARFVAE can be adapted to variable lengths, and we will investigate this and other potential benefits systematically in future work.
> > - You mention “three reasons” but list only two. Could you kindly clarify the third concern so that we can address it in full?
> >
> > We appreciate your thoroughness and look forward to your guidance on any remaining issues.

---

> > > ### Comment · Reviewer_zanJ · 2025-08-04
> > > **Follow up**
> > >
> > > I would like to apologize for the typo in my previous post. There should be two major concerns instead of three. The author also rightfully points to the corresponding section of the work that suggest the exact matching of experiment configurations. The authors prompt response is appreciated and I will update my score accordingly.

---

> > > > ### Author Response · Authors · 2025-08-05
> > > > **Response to Reviewer zanJ: Thank You**
> > > >
> > > > Dear Reviewer zanJ,
> > > >
> > > > We sincerely thank you for your careful and detailed reading of our paper and responses.
> > > >
> > > > Best regards,
> > > >
> > > > Paper 28044 Authors

---

### Official Review · Reviewer_QEJx · 2025-07-05

**Clarity:** 3
**Significance:** 2
**Originality:** 3
**Rating:** 5
**Confidence:** 4

**Summary:**

This paper introduces a new generative model for time-series forecasting, where the key idea is to combine autoregressive flow with autoencoder to achieve efficient one-step generation.

**Questions:**

- Please explain why you only compare with point forecasting models and mainly employ the point estimation metrics.
- Please add necessary related studies in studying generative time-series forecasting models and comparing point and probabilistic forecasting paradigms.
- Please add necessary experimental results to support your core claims.

**Ethical Concerns:**

["NO or VERY MINOR ethics concerns only"]

**Final Justification:**

Author's rebuttal looks good to me. Developing an efficient yet effective probabilistic forecasting method has always been a grand challenge in the time-series field, especially when forecasting horizons or variables are large. This work has made some progresses on this way.

**Limitations:**

See the weakness part.

**Quality:**

2

**Strengths And Weaknesses:**

Strengths

- Overall, I think this is a good paper, introducing an **efficient** generative model for time-series forecasting.
- Specifically, I like Section 4.2 MLP Foundation, which attempts to seperate the confounding effects of different network architectures as this paper mainly focuses on the generative forecasting framework.

Weaknesses

- My major concerns stem from the facts that although this paper proposes a generative forecasting framework, it **merely compares with recent point forecasting models** and **mainly consider point forecasting metrics**.
  - Typically, point and probabilistic forecasting studies evolve in their respective research threads and rarely mix with each other's branch. Recently, https://arxiv.org/abs/2310.07446 provided a systematic study comparing these two research branches, and according to their analyses, one major disadvantage of existing probabilistic forecasting methods lies in their inefficiency because many of them mainly considered short-term forecasting.
  - From the efficiency perspective, I think this paper has addressed a pain point of existing probabilistic models, but the experimental parts do not involve comparisons with previous probabilistic forecasting models.
  - If this paper focuses on generative models, delivering probabilistic forecasts rather than point forecasts, why the main experimental comparisons (Table 1) are presented using point metrics, such as MSE and MAE? Only in Table 2, I have found the results using the CRPS metrics, while only the results of TARFVAE have been provided. There is no performance comparisons with other models. Intuitively, the experimental comparisons in CRPS should be the core experiments to demonstrate the effectiveness of your proposed generative model.

---

> ### Author Rebuttal · Authors · 2025-07-31
>
> Thank you for your thoughtful feedback and recognition of our work. We address your concerns below:
>
> **1. Comparison scope and metrics:**
>
> We intentionally compared our generative framework with **both state-of-the-art point forecasting models (Table 1) and generative/probabilistic models (Table 3)**. We used MAE/MSE as unified metrics because:
>
> - they are widely reported across both forecasting paradigms, enabling fair cross-paradigm accuracy comparison;
>
> - they mitigate reproducibility discrepancies (e.g., mr-Diff, one of our generative baselines, only reports MAE/MSE, and lacks public code);
>
> - as noted in the paper[1] you mentioned, MAE is also adopted as a standard benchmark for probabilistic methods.
>
> Some probabilistic baselines (e.g., mr-Diff, TimeDiff) either omit CRPS for long-horizon tasks (could face extreme computational constraints) or lack public code, which makes comprehensive CRPS evaluation difficult. Nevertheless, to the best of our knowledge, ours is one of the first—and possibly the only—generative time-series forecasting works to **report MAE/MSE/CRPS at long horizons across such diverse datasets (Table 2), with full code released to facilitate any further research**. Moreover, in response to your feedback, **we add CRPS comparison following ProbTS[1] below** as it extended the evaluation of some current probabilistic methods to the long-term.
>
> **2. Related work:**
>
> We adopted state-of-the-art generative models (e.g. mrDiff, TimeDiff, TimeGrad, CSDI...) as baselines and have reviewed them in **Related Work**. We also explicitly contrasted probabilistic vs. point forecasting paradigms in **Introduction**, noting probabilistic methods further quantify uncertainty through distributional estimation. Current studies within these two paradigms, as Zhang et al.[1] indicated, differ in standardization, generation methods, and forecasting lengths. Zhang's work extended the evaluation of some probabilistic methods to the long-term and conducted a comprehensive comparison between the two paradigms. We will include the suggested paper[1] to enrich this discussion in later version. We've also compared our results with Zhang's, as shown in the next response.
>
> **3. CRPS results:**
>
> Following your suggestion, we benchmark our model against baselines in ProbTS[1] (including both point and generative/probabilistic models) using the identical CRPS protocol. We produce this new CRPS without retraining, and the baseline results are taken from Table 10 in [1]. **Results below show our TARFVAE still outperforms all probabilistic and point competitors in CRPS except for ranking 2nd in one case, demonstrating the effectiveness of our generative framework.**
>
> **Table: CRPS results following identical protocol of ProbTS. Baseline results are taken from ProbTS. Bold numbers indicate the best performance.**
> | Dataset | Horizon | TARFVAE (ours)| Other Top Performance |
> |:---|:---|:---|:---|
> | ETTm1 | 96 | **0.222** | 0.236(CSDI) |
> |  | 192 | **0.244** | 0.291(CSDI) |
> |  | 336 | **0.261** | 0.322(CSDI) |
> |  | 720 | **0.291** | 0.353(PatchTST) |
> | ETTm2 | 96 | **0.110** | 0.115(CSDI) |
> |  | 192 | **0.131** | 0.147(CSDI) |
> |  | 336 | **0.150** | 0.176(PatchTST) |
> |  | 720 | **0.171** | 0.205(PatchTST) |
> | ETTh1 | 96 | **0.254** | 0.321(iTransformer) |
> |  | 192 | **0.278** | 0.359(iTransformer & PatchTST) |
> |  | 336 | **0.301** | 0.384(PatchTST) |
> |  | 720 | **0.318** | 0.397(PatchTST) |
> | ETTh2 | 96 | **0.150** | 0.164(CSDI) |
> |  | 192 | **0.173** | 0.201(PatchTST) |
> |  | 336 | **0.185** | 0.240(PatchTST) |
> |  | 720 | **0.190** | 0.252(PatchTST) |
> | Electricity | 96 | **0.068** | 0.086(PatchTST) |
> |  | 192 | **0.077** | 0.092(PatchTST) |
> |  | 336 | **0.081** | 0.099(GRU NVP) |
> |  | 720 | **0.087** | 0.108(TimeGrad) |
> | Weather | 96 | **0.065** | 0.068(CSDI) |
> |  | 192 | 0.070 | **0.068(CSDI)** |
> |  | 336 | **0.074** | 0.083(CSDI) |
> |  | 720 | **0.080** | 0.087(CSDI) |
> | Exchange | 96 | **0.019** | 0.023(PatchTST) |
> |  | 192 | **0.028** | 0.034(PatchTST) |
> |  | 336 | **0.041** | 0.048(iTransformer & PatchTST & DLinear) |
> |  | 720 | **0.068** | 0.072(PatchTST) |
>
> References:
>
> [1] Zhang et al. "ProbTS: Benchmarking Point and Distributional Forecasting across Diverse Prediction Horizons". NIPS 2024.

---

> > ### Comment · Reviewer_QEJx · 2025-08-04
> > **Thank you for your response**
> >
> > It is great to see these supplemented experimental results. The reported CRPS scores of long-term probabilistic forecasting are pretty good.So I raised the score.
> >
> > Besides, I would like to suggest further comparing with existing probabilistic forecasting methods in the efficiency aspect as your method seems much more efficient. As far as I know, some of existing probabilistic methods ran into computation obstacles when either forecasting horizons or time-series variables increase.

---

> > > ### Author Response · Authors · 2025-08-04
> > > **Thank you for the feedback**
> > >
> > > We sincerely appreciate your positive feedback and constructive suggestion. We fully agree on the importance of computational efficiency for long-term probabilistic forecasting. In later version, we will include detailed analyses of computational cost and complexity. We are also actively preparing a comprehensive benchmark; given the scope required for faithfully reproducing existing probabilistic methods, the full evaluation will be incorporated in the upcoming version and open-sourced to facilitate further research.

---

### Note · Authors · 2025-08-12

We sincerely thank all reviewers and the AC for their constructive engagement. We are encouraged that reviewers acknowledged our method’s **SOTA performance and exceptional computational efficiency** (QEJx, zanJ, 3TPd, EWqS), **praised its elegant and creative design** (3TPd, EWqS), and **commended the paper’s rich yet accessible technical exposition** (zanJ) alongside its **overall quality** (QEJx).

Here, we want to highlight the novelty and contributions of our proposed method as follows:

- **First efficient one-step long-horizon generative time-series forecasting**: By unifying flow and VAE, we enable full-horizon sampling in one forward pass—addressing computational bottlenecks in diffusion/autoregressive models. We maintain O(N²) complexity—matching iTransformer and DUET—thus achieving efficiency comparable to these deterministic models and up to 100× faster inference than diffusion baselines.

- **TARFLOW for expressive posterior approximation**: We introduce a unidirectional Transformer-based autoregressive flow (TARFLOW) to break VAE’s Gaussian posterior constraint, yielding a richer latent space that empowers a lightweight MLP backbone to reach SOTA performance.

- **Comprehensive and reproducible evaluation**: We report MSE, MAE, and CRPS across eight widely used benchmark datasets and horizons up to 720, outperforming both deterministic and probabilistic competitors, while releasing full code to ensure reproducibility.

We summarized our major responses to reviewers as follows:

- We added CRPS comparisons against baselines in ProbTS (including both point and generative/probabilistic models) using the identical CRPS protocol. (QEJx)

- We provided mean-over-samples metrics and per-timestep MSE. (3TPd)

- We replaced the MLP with iTransformer to demonstrate architecture-agnostic performance. (zanJ, EWqS)

- We elaborated on the rationale for selecting C-VAE and its operational efficacy. (3TPd)

All reviewer concerns have been comprehensively resolved in the detailed responses below their reviews. Please kindly check them out. Thank you and please feel free to ask any further questions.

---

### Decision · Program_Chairs · 2025-09-17

**Decision:**

Accept (poster)

**Comment:**

The submission presents a time series generation approach highlighting generation efficiency for longer horizon. This is achieved by a one-step, concurrent VAE generation process, while from the performance part, an autoregressive normalizing flow encoder is used to lift the usual conditional Gaussian model for a more precise capture of the posterior distribution and a more informative latent code reflecting the given condition. The proposed approach has been shown to outperform prevalent existing approaches in forecasting quality and efficiency.

Reviewers and myself appreciate the investigation of this combination of models and the resulting preferred performance. The technical choices seem solid and well-motivated. Reviewers also raised some major insufficiencies, regarding which the authors also posted their responses:
* Evaluation baselines and metrics lacking generative/probabilistic versions, which are expected due to the generative nature of the proposed method. The authors provided one case in the rebuttal which seems supportive, while argued that more metrics and baselines are less ready for running a comparison. This seems a reasonable situation.
* Limited technical novelty. The work indeed closely follows the framework by Rezende et al. (ICML 2015). I suppose the effort to develop appropriate architectures and temporal structure design within this framework for improving time series generation could still be worth a contribution. (Regarding authors rebuttal, I suppose the original work by Rezende et al. also "avoids the costly bijective inversion at inference", as its generation process follows VAE but not the flow.)
* Capability to generate varying-length time series. This may be a common asking for one-step time-series generation approaches. Though it is acceptable to first focus on fixed-length generation performance, I still hope the authors could include such a demonstration.

Overall, I suppose the contribution of this paper outweighs its current limitations. I hence recommend an acceptance.